

# Diversity and distribution of air-breathing sea slug genus *Peronia* Fleming, 1822 (Gastropoda: Onchidiidae) in southern Japanese waters

Iori Mizukami[1], Chloé Julie Loïs Fourreau[1], Sakine Matsuo[1] and James Davis Reimer[1,2]

[1] Molecular Invertebrate of Systematics and Ecology (MISE) Lab, Graduate School of Engineering and Science, University of the Ryukyus, Nishihara, Okinawa, Japan
[2] Tropical Biosphere Research Center, University of the Ryukyus, Nishihara, Okinawa, Japan

## ABSTRACT

Species of the genus *Peronia* Fleming, 1822, are air-breathing onchidiid sea slugs that inhabit intertidal reef flats of temperate to tropical zones. In the Ryukyu Islands of southern subtropical Japan, *Peronia* species are a traditional food source for local people. To date, there have been three species recorded around Okinawajima Island; *P. verruculata* and *P. peronii*, along with recently described *P. okinawensis*, which was described as possibly endemic to Okinawajima Island. This study aimed to map the distribution ranges of these three *Peronia* species within the Ryukyu Islands using molecular analyses in order to understand the specific distribution of each species. Since *Peronia* species are generally indistinguishable by gross external morphology, a DNA barcoding approach was employed to identify specimens. The molecular data showed that there are four species present in the Ryukyu Islands. *P. verruculata* (unit #1 *sensu* Dayrat et al., 2020) was dominant at almost all locations, while *P. peronii* was present in much lower numbers than *P. verruculata*, but found across a relatively wide range in the Ryukyu Islands. We newly record *P. okinawensis* and *P. setoensis* from Amami Oshima Island and from several places around Okinawajima Island, and also identified high levels of genetic variation within *P. setoensis*. *Peronia okinawensis* and *P. setoensis* have been thought to be endemic to Okinawajima Island and to Honshu, mainland Japan, respectively. However, as both species were observed around Okinawajima and Amami Oshima islands, other islands of the Ryukyus are also likely to harbor these species, and their distribution ranges are wider than previously thought. Based on the results from molecular analyses, we provide general descriptions of each species. Sizes of specimens were consistently smaller for *P. setoensis* and relatively larger for *P. peronii* specimens. On the other hand, *P. verruculata* and *P. okinawensis* had similar size ranges, but *P. okinawensis* had comparatively much more distinct papillae. This study revealed that the Ryukyu Islands are the only region currently known with four sympatric *Peronia* species, and this work provides a basis for future research on these *Peronia* species throughout the northwest Pacific Ocean, representing the first step in more effective management of the local *Peronia* fisheries in the Ryukyu Islands.

Corresponding author
Iori Mizukami,
iori.mizukami@my.jcu.edu.au

## INTRODUCTION

Molecular analyses have become widely used among biodiversity researchers in recent years, and morphologically indistinguishable cryptic species have been revealed in many different taxa (*Saez et al., 2003*). There have been many recent taxonomic studies on a variety of Mollusca groups including molecular datasets that have successfully applied to species delineation (*Kano et al., 2016*; *Soong, Wilson & Reimer, 2020*; *Feliciano et al., 2021*; *Soong et al., 2022*). Furthermore, molecular analyses are not only useful to make taxonomic and identification conclusions regarding cryptic species, but also are critical for understanding the distribution ranges of species (*Handayani et al., 2020*). Such data are especially crucial when the cryptic species are few in numbers of individuals, or limited to particular sites, and have little differences on the external gross morphology between existing species (*Westram et al., 2011*). Moreover, the molecular data and distribution information of such cryptic species are useful for assessing populations and their habitat preferences (*Brown, 1984*). Thus, DNA barcoding is a method that can clarify the distribution ranges and inform subsequent conservation of species.

Species of the genus *Peronia* Fleming, 1822 are air-breathing onchidiid sea slugs that inhabit intertidal reef flats especially on hard substrates such as coral rubbles, rocks, artificial structures (*e.g.*, tetrapods (*Masucci, Acierno & Reimer, 2019*), and concrete walls (*Dayrat, 2009*; *Goulding et al., 2021*)). *Peronia* spp. are distributed in the entire tropical and the subtropical West Indo-Pacific, from latitudes between South Africa to Hawaii (*Wu et al., 2010*; *Dayrat et al., 2020*), and they are observed frequently in temperate to subtropical regions of Japan (*Katagiri, Fujimoto & Katagiri, 1983*; *Takagi et al., 2019*). One of the characteristics of onchidiid sea slugs is a shell-less notum with papillae, which contains several dorsal eyes (*Yanase & Sakamoto, 1965*; *Okuno, Katagiri & Fujimoto, 1976*; *Katagiri, Katagiri & Fujimoto, 1981*). These dorsal eyes work as photoreceptors and they are a scientifically important model in neurophysiology studies (*e.g.*, *Nishi & Gotow, 1992*; *Katagiri & Katagiri, 2008*; *Gotow & Nishi, 2009*).

Moreover, in the Ryukyu Islands (also known as the Nansei Islands) of southern subtropical Japan, *Peronia* species are a traditional food source due to their abundance along the shore (*Kubo & Kurozumi, 1995*). This *Peronia* fishery is well-known as a unique small fishery on some islands in Okinawa and Kagoshima Prefectures; particularly on Ie, Iheya, Izena, Amami Oshima, Tokunoshima and Okinoerabujima islands (*Masaoka & Magara, 2021*). *Peronia* spp. are collected mostly in fresh form and prepared as sashimi or stir-fried with miso paste. Nevertheless, since only a few molecular studies on *Peronia* species have been performed in Japan and Ryukyu Islands, these fisheries may not completely understand which species they are harvesting. From this situation, it is clear there is a need to assess *Peronia* spp. diversity and ranges in southern Japan.

*Peronia verruculata* (Cuvier, 1830) was believed to be the only species in Japan for many years. However, in 2007, researchers noticed that there was more than one *Peronia* species

existing sympatrically in Japan (*Katagiri & Katagiri, 2007*; *Ueshima, 2007*). Although the morphological differences were subtle, it was concluded that *Peronia* species could be distinguished *via* external morphology, and researchers identified relatively small individuals as a species (tentatively called *Peronia* sp. or "mini awamochi" in Japanese) different from larger individuals (*Peronia verruculata* or "iso awamochi") (*Katagiri & Katagiri, 2007*). This smaller *Peronia* sp. turned out to be an undescribed species that was formally described as *P. setoensis* Dayrat & Goulding, 2020, and thus to date there have been a total of four species recorded in Japan; *P. verruculata*, *P. peronii* (Cuvier, 1804), *P. okinawensis* Dayrat & Goulding, 2020, and *P. setoensis*. The "iso awamochi", *P. verruculata*, has been recorded in many locations in Japan southwards from the Bōsō Peninsula, Chiba Prefecture (ca. 35°N), but the distribution of *P. setoensis* remains unclear, particularly in the Ryukyu Islands and it is not known if *P. setoensis* is endemic to Honshu, mainland Japan (*Ueshima, 2007*). Moreover, there have been three species recorded in the Ryukyu Islands; *P. verruculata* and *P. peronii*, along with recently described *P. okinawensis* (*Dayrat et al., 2020*). *P. okinawensis* is possibly endemic to Okinawajima Island, as the specimens *Dayrat et al. (2020)* used in the formal description were collected in 2004 from only one location in northern Okinawa Island. Since this description, there have been no other records of *P. okinawensis* published, and the distribution range of *P. okinawensis* remains unknown.

Furthermore, the distribution of *Peronia* species in southern Japan were reported by *Takagi et al. (2019)*, who applied DNA barcoding on several onchiid sea slug specimens from the southern part of Japan including the Okinawajima Islands. Their molecular analyses confirmed that *P. verruculata* was present all over Okinawajima Island north to Kagoshima Prefecture, the southernmost part of mainland Japan. However, two larger *Peronia* specimens showed clear genetic differences from other *P. verruculata* specimens and these were only found at a specific site on the eastern side of Okinawajima Island. *Takagi et al. (2019)* did not identify these individuals due to low numbers of specimens and restricted studied sites, but the characteristics of the specimens and sequences matched with those reported previously for *P. peronii*. Thus, *Takagi et al. (2019)* concluded that animal size could be useful for the identification of *Peronia* species, but mentioned more detailed analyses would be required on a wider variety of specimens and sites to confirm this.

Therefore, this study aimed to better confirm the distribution ranges of *Peronia* species within the Ryukyu Islands. We used analyses of mitochondrial DNA marker sequences (COI and 16S rDNA) in order to achieve three main objectives:

1. To determine the number of *Peronia* species that exist in the Ryukyu Islands and better understand their distribution ranges. Understanding distributions is critical for informing *Peronia* fisheries and to aid in conservation given the ongoing coastal development in these islands (*Masucci & Reimer, 2019*).
2. To investigate the phylogenetic relationships of *Peronia* species in the Ryukyu Islands, as well as to assess population genetic differences of *Peronia* species within the Ryukyu Islands *via* haplotype network analyses.

3. To better characterize the external morphological characteristics of *Peronia* spp. in the Ryukyu Islands to better inform future studies.

## MATERIALS AND METHODS

### Specimen collection and external morphological analyses

A total of 87 *Peronia* specimens were collected *via* reef walking in the intertidal zones at low tide from several places in southern Japan, mainly around the Okinawajima Island, Iriomote Island, Amami Oshima Island, and in Kochi Prefecture, mainland Japan from August 2020 to December 2021 (Table S1; Fig. 1). Specimens were collected randomly depending on the differences of obvious external morphology such as their size. Once found, all specimens were photographed with their surroundings *in situ*, then the specimens were brought back to the laboratory. Although this study did not conduct detailed internal morphological analyses, the sizes of all specimens were measured when animals were walking (length/width to the nearest mm). At the same time, all specimens were photographed for analyzing external characteristics such as colors and patterns on the dorsal and ventral sides of animals. Subsequently, the specimens were preserved in 99.5% ethanol for further molecular analyses. All specimens collected in this study are currently kept in a collection by the first author, and will be deposited in the Fujukan University of the Ryukyus Museum in the future.

No collection permits were needed for *Peronia* collection in Okinawa or Kochi Prefectures, and specimens from Kagoshima were collected on Kagoshima Prefecture permit Shirei Oshima Rinsui #2006-7.

### DNA extraction, PCR amplification and sequencing

DNA was extracted from mantle tissues of each preserved specimen using Qiagen DNeasy Blood and Tissue Kit (Qiagen, Tokyo, Japan) following the manufacturer's protocol. Two mitochondrial genes; cytochrome *c* oxidase subunit I (COI) and 16S ribosomal DNA (16S rDNA) were amplified using polymerase chain reaction (PCR). For each marker, PCR amplifications were carried out in a 20 μL reaction volume with 10 μL of Hot Start Taq Plus Master Mix Kit (Qiagen, Tokyo, Japan), 7 μL of H2O, 1 μL of each primer and 1 μL of extracted DNA template. Universal COI (*Folmer et al., 1994*) and 16S primers (*Palumbi, 1996*) were used for amplifications. The following thermo-cycler conditions were utilized to amplify a 658 base pair (bp) region of COI; an initial denaturation at 95 °C for 5 min, further denaturation at 94 °C for 30 s, annealing at 46 °C for 30 s, and extension at 72 °C for 1 min for 39 cycles, and the final elongation at 72 °C for 5 min. To amplify 560 bp of 16S rRNA, conditions were; an initial denaturation at 94 °C for 5 min, 39 cycles of further denaturation at 94 °C for 30 s, annealing at 52 °C for 30 s, and extension at 72 °C for 1 min, with the final elongation at 72 °C for 5 min (*Johnson & Gosliner, 2012*; *Soong, Wilson & Reimer, 2020*). Successfully amplified PCR products were cleaned and purified with Exonuclease I–Shrimp Alkaline Phosphatase (ExoSAP). Cleaned PCR products were sequenced in both directions at FASMAC (Kanagawa, Japan).

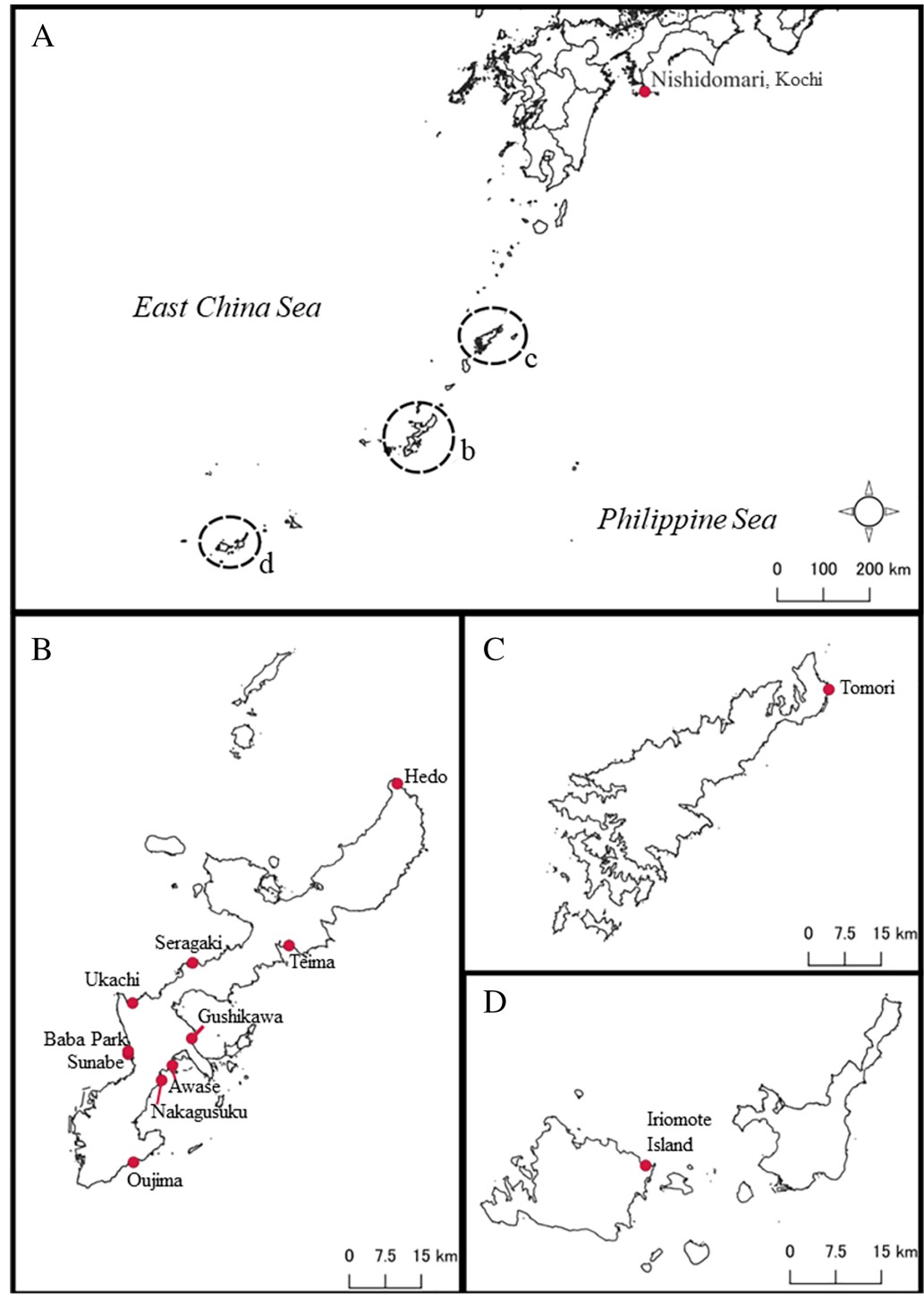

**Figure 1 Map of sampling locations of *Peronia* species in this study in southern Japan.** (A) Map of southern Japanese waters. Close-up maps of (B) Okinawajima Island, (C) Amami Oshima Island, and (D) the Yaeyama Islands including Iriomote Island.

## Phylogenetic analyses

Newly obtained sequences were aligned, trimmed, and edited using MUSCLE (*Edgar, 2004*) within Geneious version 8.1.9 (*Kearse et al., 2012*). As the new sequences were confirmed using BLAST to match to existing GenBank sequences of *P. verruculata*, *P. peronii*, *P. okinawensis* and *P. setoensis* for both COI and 16S, additional sequences of these four species and outgroup onchiid sp. (*Wallaconchis ater*) from GenBank were retrieved and used in phylogenetic analyses (Table S1). Consensus sequences and retrieved GenBank sequences were trimmed to the same length. The alignments of COI and 16S were concatenated using SequenceMatrix version 1.8 (*Vaidya, Johman & Meier, 2011*) for phylogenetic analyses.

Phylogenetic trees were reconstructed for each marker respectively and for the concatenated dataset. Two types of phylogenetic trees were created; Bayesian inference (BI) and maximum likelihood (ML) analyses. The best fit model of sequence evolution was decided using find best DNA/protein models within MEGA X (*Kumar et al., 2018*) in default settings. The best model was chosen based on AIC criterion (*Akaike, 1974*). Models for COI and 16S partitions were GTR+G (*Tavaré, 1986*) and HKY+G (*Hasegawa, Kishino & Yano, 1985*), respectively. BI analyses were performed in MrBayes version 3.2.7 (*Ronquist & Huelsenbeck, 2003*) using concatenated dataset. Bayesian Markov chains were conducted with two runs of three chains each for $2 \times 10^6$ generations which were sampled every 100 generations. The burn-in was set to 25%. ML analyses were constructed using RaxML-NG version 1.0.2. (*Kozlov et al., 2019*) using the same concatenated dataset with model partitions. The command all was employed to conduct tree search and 1,000 bootstrap repetitions. PhyML 3.0 (*Guindon et al., 2010*) was performed with default setting and SMS automated model selection (*Lefort, Longueville & Gascuel, 2017*) on the concatenated dataset to confirm the results of RAxML-NG. For each single gene marker, ML analyses were conducted using MEGA-X (*Kumar et al., 2018*) using 1,000 bootstrap replications. Posterior probabilities (PP) $\geq 0.9$ and Bootstrap values $\geq 75\%$ were considered as significant. FigTree v1.4.3 (*Rambaut, 2016*) was used for final visualization and coloring of the trees.

## Species delimitation analyses

For species delimitation, a pairwise distance matrix was calculated using the raw distance model for the COI alignment with the package *ape* (version 5.5., *Paradis & Schliep, 2019*) and visualized in R version 4.1.0. The Automatic Barcode Gap Discovery (ABGD) algorithm (*Puillandre et al., 2011*) was also carried out for considering species boundaries as well as examining intraspecific distances (https://bioinfo.mnhn.fr/abi/public/abgd/abgdweb.html). The analyses settings were set as default using the simple distance model. All analyses were performed with the same COI alignment that was used in the ML tree inference.

## Haplotype analyses

For visualizing the correspondence between each site and haplotype of *Peronia* species, a haplotype network was constructed using the mitochondrial COI and 16S datasets of 81

and 75 sequences, respectively. The alignments were loaded into R version 4.1.0 and visualized with package *pegas* (*Paradis, 2010*). The statistical parsimony was calculated using *Templeton & Sing (1993)*'s method, which computes the estimated maximum number of base pair changes among haplotypes as a result of single mutations. It computes a cladogram with >95% probability linkages and constructs groups of haplotypes differing by one base pair change, then two changes, three, and so on (*Templeton & Sing, 1993*).

## Morphological data analyses

Average lengths of animals were visualized using R version 4.1.0 for comparing average lengths between species as well as between collection sites. Moreover, one-way ANOVA was conducted using *rstatix* package 0.7.0 (*Kassambara, 2021*) for examining correlations between average lengths of each species, and between average lengths and collection site. Tukey's HSD test was run after an ANOVA analysis to find out which specific group's mean were different when comparing each group. Compact letter displays for all analyses were performed on R using *multcomp* package v1.4-16 (*Hothorn, Bretz & Westfall, 2008*).

## RESULTS

From a total of 87 specimens, 81 COI and 75 16S sequences were successfully retrieved, and in total 84 specimens were identified to species level. All new COI and 16S sequences generated for this study were deposited in GenBank (Table S1). The generated alignments were trimmed to 111 taxa of 548 bp in length for COI, and 100 taxa of 325 bp in length for 16S rDNA.

### Phylogenetic analyses

Bayesian inference (BI) analyses of concatenated dataset (COI+16S) (Fig. 2) and Maximum likelihood (ML) of single genes of COI and 16S rDNA (Figs. S1 and S2) showed similar topologies and recovered the same clades among our *Peronia* specimens. BI analyses revealed two clades within the specimens from Japan/Ryukyu Islands: a *P. verruculata* & *P. setoensis* clade (PP = 1) and a *P. peronii* & *P. okinawensis* (PP = 1) clade. Within these clades, the monophylies of both two species were well-supported, except for the five units of *P. verruculata*, but all *P. verruculata* from the Ryukyu Islands belonged to *P. verruculata* unit #1 (*sensu Dayrat et al., 2020*). Within the clade of *P. setoensis*, there were small subclades that were divided depending on sampling location. All *P. peronii* specimens in this study and retrieved sequences from Pacific regions (Guam, Papua New Guinea and Okinawajima Island) belonged to the same clade, but a sequence from a Mauritius specimen from GenBank (GB18) was retrieved as a separate singleton. All *P. okinawensis* specimens in this study and the holotype from Hedo, Okinawajima Island in 2004 were in the same clade. ML analyses of concatenated dataset (COI+16S) were performed (not shown), but as there were only subtle differences in the topology of the *P. verruculata* & *P. setoenisis* clade from other phylogenetic trees, we decided to focus on results of BI analyses of the concatenated dataset.

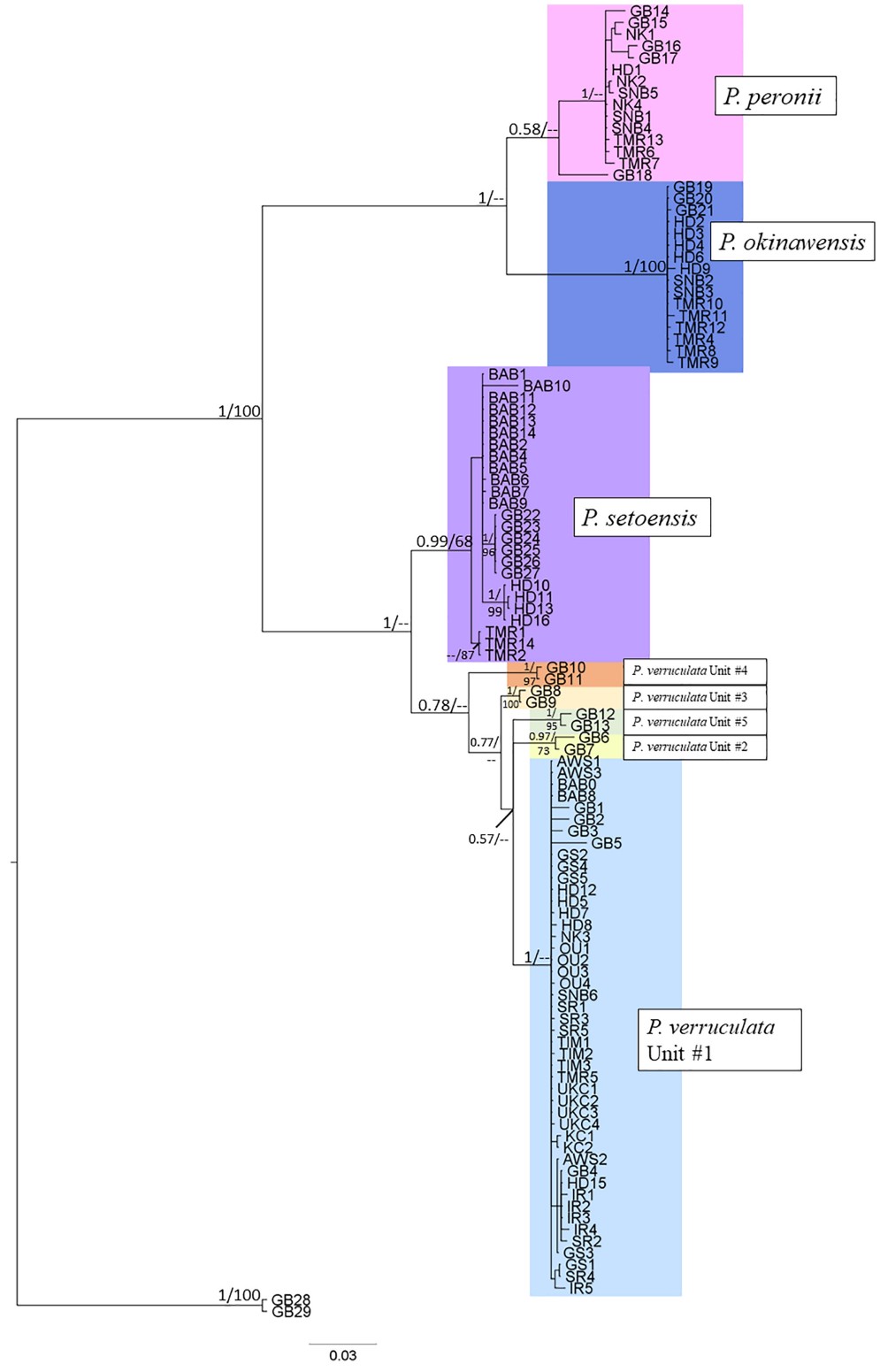

**Figure 2 Bayesian Inference (BI) consensus tree based on concatenated mitochondrial COI and 16S gene markers.** Posterior probabilities from the BI analysis and the bootstrap values from maximum likelihood (ML) analysis are listed on the branches (PP/bootstrap value). The numbers indicated on the branch are PP ≥ 0.5 and bootstrap value ≥50.

## Species delimitation

The species delimitation with ABGD analysis of the COI sequence data using the uncorrected pairwise distance model supported four species among specimens collected in this study. The pairwise COI distances among Japanese *Peronia* specimens are displayed in Fig. 3 and Table 1. The minimum genetic distances separating *Peronia* ABGD species-groups was 8.6% (between *P. verruculata* and *P. setoensis*) and the maximum was 18.8% (between *P. okinawensis* and *P. setoensis*). The *p*-COI distances suggested that *P. setoensis* from the Ryukyu Islands were not a distinct species from Wakayama Prefecture specimens, and the genetic distance between the Ryukyu Islands specimens and Wakayama specimens ranged from 0.9% to 2.4%. However, the *p*-COI distances among *P. setoensis* specimens in this study were 0% to 3.5%, the largest genetic distances among the four species examined (Table 1).

## Haplotype analyses

The haplotype network using COI and 16S rDNA sequence data showed a total of 34 haplotypes for COI and 20 haplotypes for 16S among our *Peronia* specimens (Fig. 4; Tables S2 and S3). For each species, there were 17/8 (COI/16S) haplotypes for *P. verruculata*, 6/5 for *P. peronii*, 5/3 for *P. okinawensis*, and 6/4 for *P. setoensis*. Although the number of collected specimens was the lowest in *P. peronii*, *P. okinawensis* had less haplotypes than *P. peronii*. There was one very common haplotype of *P. verruculata* in both markers found from all around Okinawajima Island, as well as from Iriomote Island and Amami Oshima Island in the 16S marker. Moreover, *P. peronii* and *P. okinawensis* had similar patterns in that one common haplotype was obtained from several collection sites and several less frequent haplotypes with one to four base pair changes from the common haplotype were seen in both markers (Fig. 4). On the other hand, haplotypes of *P. setoensis* showed clear separation depending on collection sites as also shown in phylogenetic analyses (Figs. 2 and 4). There were 19 to 20 base pair changes in COI and six to nine base pair changes in 16S between each location (Baba Park, Hedo, Amami (Tomori)) for *P. setoensis* (Fig. 4).

## Morphological analyses

The largest and smallest individuals among all collected specimens were; *P. verruculata* (largest: length 70 mm/width 37 mm, smallest: 7 mm/4 mm, length average: 37.56 ± 13.69 mm, width average: 20.62 ± 9.65 mm; *n* = 41), *P. peronii* (110/100, 14/8, 78.6 ± 32.49 mm, 53.75 ± 31.43 mm; *n* = 10), *P. okinawensis* (60/40, 20/13, 45.23 ± 13.82 mm, 25.36 ± 8.33 mm; *n* = 13) and *P. setoensis* (28/16; 5/4, 14.65 ± 7.09 mm, 9.65 ± 5.17 mm; *n* = 20) (Table S1, Fig. 5A). There were wide variations in sizes within the same species, however the average size of *P. peronii* were generally bigger than the other species (F $(2,61) = [21.449]$, $p = 8.83 \times 10^{-8}$), and *P. setoensis* had smaller sizes on average (F $(2,71) = [31.579]$, $p = 1.55 \times 10^{-10}$) based on results of one-way ANOVA. Tukey's HSD Test for multiple comparisons found that the average length was significantly different between *P. setoensis* and *P. verruculata*, *P. peronii*, *P. okinawensis*, as well as between *P. peronii* and *P. verruculata*, and *P. peronii* and *P. okinawensis* (Fig. 5A). *P. verruculata* and *P. okinawensis* had similar size ranges, but *P. okinawensis* was slightly bigger than

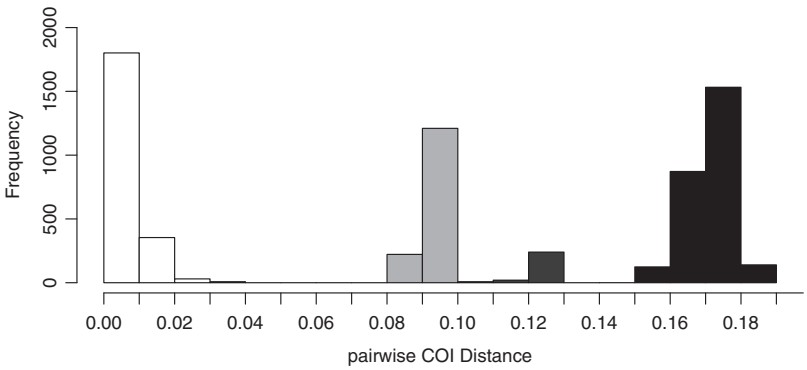

**Figure 3 Results of *p*-COI distances on ABGD analysis based on COI sequence data obtained in this study.** White-colored bars indicate intra-specific genetic distances, grey minimum inter-specific genetic distances, and black maximum inter-specific genetic distances.

**Table 1 Pairwise genetic distances of mitochondrial COI sequences between four *Peronia* species in the Ryukyu Islands (in percentage).** Percentage ranges indicate minimum to maximum genetic distances

| Species | 1 | 2 | 3 | 4 |
|---|---|---|---|---|
| 1 *P. verruculata* | 0–1.6 | | | |
| 2 *P. setoensis* | 8.6–10 | 0–3.5 | | |
| 3 *P. peronii* | 16.6–17.7 | 15.3–17.7 | 0–1.1 | |
| 4 *P. okinawensis* | 17–18.4 | 16.8–18.8 | 11.7–12.6 | 0–0.9 |

*P. verruculata* (Figs. 5Ba. and 5Bd.). The average body sizes depending on collection sites are shown in Fig. 6. A one-way ANOVA was performed to check whether collection sites affected specimen sizes, and there was a significant difference in specimen sizes between Kochi and Iriomote Island and between Kochi and Baba Park of *P. verruculata* ($F(9,28) = [2.566]$, $p = 0.0272$) as well as between Baba Park and Hedo, and between Baba Park and Amami for *P. setoensis* ($F(2,17) = [35.26]$, $p = 8.94 \times 10^{-7}$) (Fig. 6). There were no significant differences between the location and specimen sizes in *P. peronii* and *P. okinawensis*.

The pictures of live animals showed color variations within each species and external gross morphological differences between the species (Figs. 7–10). *Peronia* specimens investigated in this study had dorsal colors that were roughly classified into light to dark brown, grey with green-ish colors and sometimes marbled with two colors. Their papillae colors also varied the same as body colors with different colors. The ventral side was white to yellowish light brown and some specimens showed green colors on the inner to the edge of the mantle. *P. verruculata* specimens had the most color variation patterns among four species (Figs. 7A–7F). On the other hand, all *P. setoensis* specimens had green-ish grey color with a little light brown (Fig. 8). *P. peronii* had the lowest specimen numbers, but they also had similar color morphotype variations as seen in *P. verruculata*. However, colors of *P. peronii* were much darker in general than in *P. verruculata* (Fig. 9).

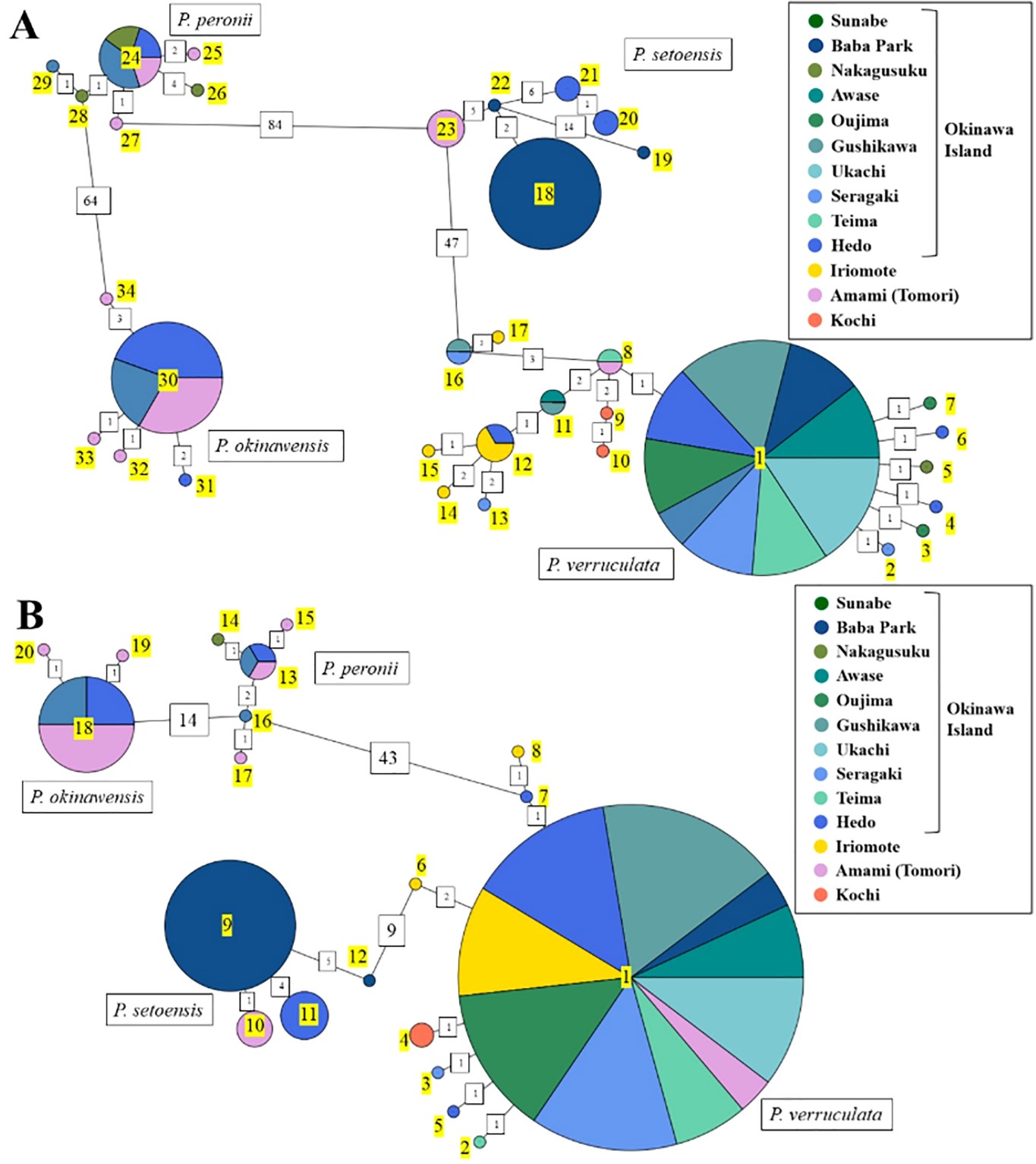

**Figure 4 Haplotype network based on (A) COI sequences and (B) 16S rDNA sequence data.** Colors represent the collection sites in this study as in the legend. Haplotype numbers in yellow. Green colors represent the east coast and blue colors represent the west coast of Okinawajima Island. Haplotypes COI: 1–17, 16S: 1–8 = *P. verruculata*, 18–23, 9–12 = *P. setoensis*, 24–29, 13–17 = *P. peronii*, 30–34, 18–20 = *P. okinawensis*.

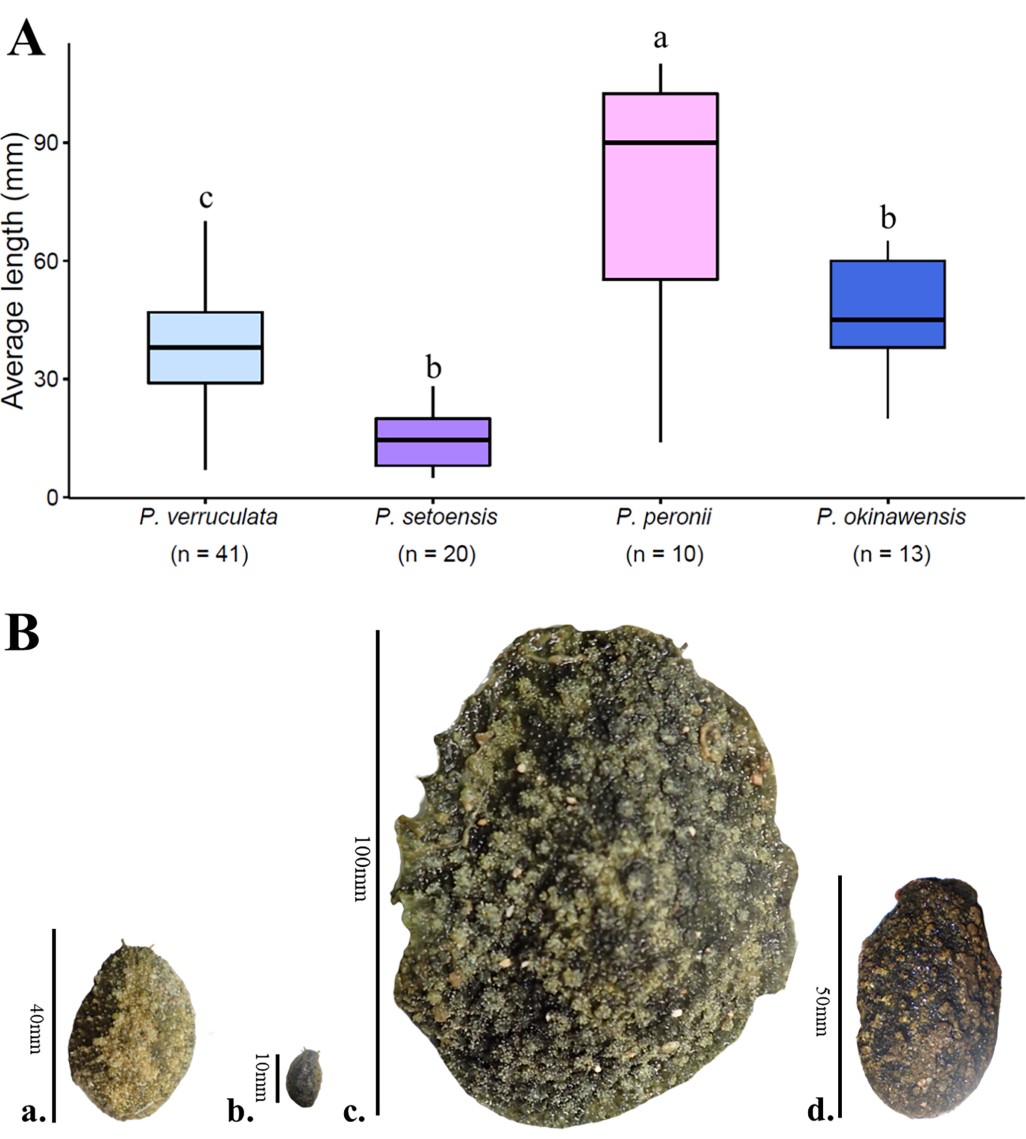

**Figure 5 Average specimen lengths of four species in this study with (A) box plot and (B) four *Peronia* images with scales.** (A) *n* = total numbers of collected specimens and small letters in the graph indicates significance differences at the 0.05 probability. (B) (a.) *P. verruculata*: HD5 (36/26); Okinawajima Island, Hedo, Uza Beach (26°51′57.1″N, 128°15′47.1″E), (b.) *P. setoensis*; BAB14 (13/9); Okinawajima Island, Sunabe, Baba Park (26°20′04.9″N, 127°44′35.9″E), (c.) *P. peronii*: HD1 (length: 105 mm/ width: 81 mm), (d.) *P. okinawensis*: TMR12 (50/31); Amami Oshima Island, Tomori Beach (28°27′43.8″N, 129°43′18.1″E).

*P. okinawensis* generally had green-ish darker grey to brown body color (Fig. 10), except for one specimen from Hedo (Fig. 10C) that had a plain light brown dorsal color. Moreover, from analyses of photographs of living animals *in situ*, the papillae were comparatively more prominent in *P. okinawensis* and *P. peronii* and relatively smoother in *P. setoensis*. Especially, all *P. okinawensis* specimens in this study were roughly classified into two distinct types of papillae shape; seven specimens had papillae similar to "goya", or bitter melon-like in shape (Figs. 10A–10C), and six specimens had much more dense papillae that were tiny and chestnut shell-like in shape (Figs. 10D–10F).

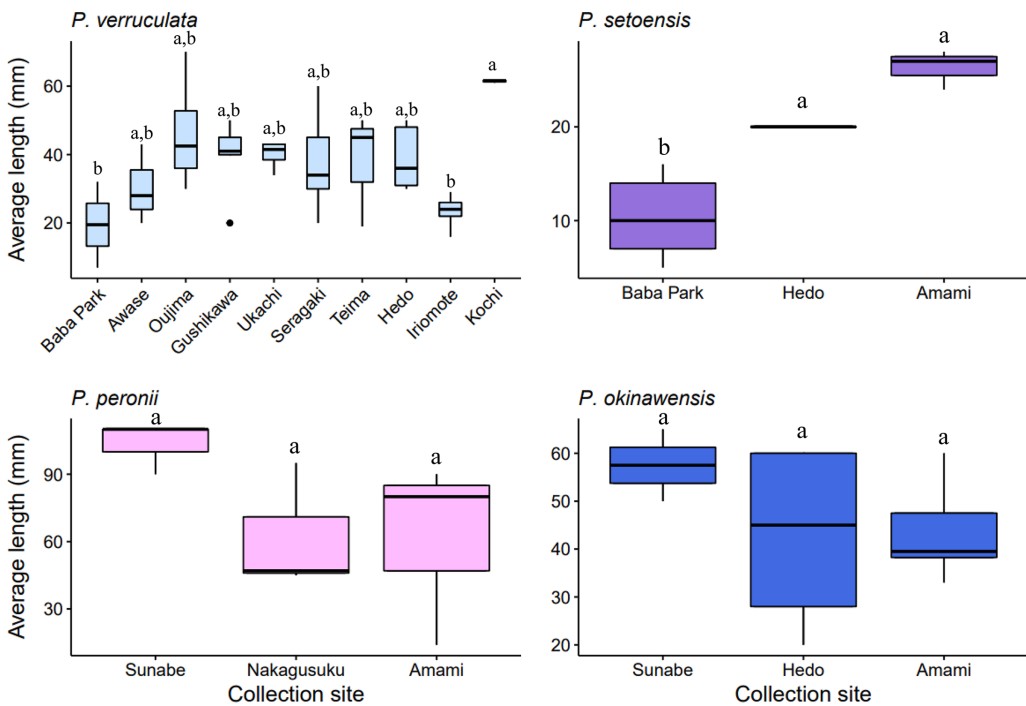

**Figure 6 Box plot of the average specimen lengths of four *Peronia* species by collection sites.** Locations with less than two specimens not included. Small letters in the graphs indicate significance differences at the 0.05 probability level.

## Species distributions

*Peronia verruculata* unit #1 was found at all study sites and was the most abundant species at many sites (Fig. 11). *Peronia peronii* were observed from several places around Okinawajima Island and at the northern part of Amami Oshima Island (Fig. 11). *P. peronii* was not found in the Sakishima Islands (Fig. 11). *Peronia okinawensis* and *P. setoensis* had similar distribution ranges; the western coast of Okinawajima Island as well as northern Amami Oshima Island (Fig. 11).

## DISCUSSION

The results of this study showed that the Ryukyu Islands harbor four *Peronia* species, while it was previously thought there were only three species in this region (*Dayrat et al., 2020*). The Ryukyu Islands are the only region globally currently known with four sympatric *Peronia* species, as other regions have only one or two species co-existing (*Goulding et al., 2021*). The three species that were already recorded from this region were *P. verruculata, P. peronii* and *P. okinawensis*, while the newly recorded species was *P. setoensis*, which was previously thought to be endemic to mainland Japan. The *Peronia* species composition of Amami Oshima Island was shown to be the same as that of Okinawajima Island, with all four species recorded from both islands. However, unlike Okinawajima Island where *Peronia* species were observed at all sites investigated, *Peronia* species were mostly abundant in the northern part of Amami Oshima Island from observations by colleagues.

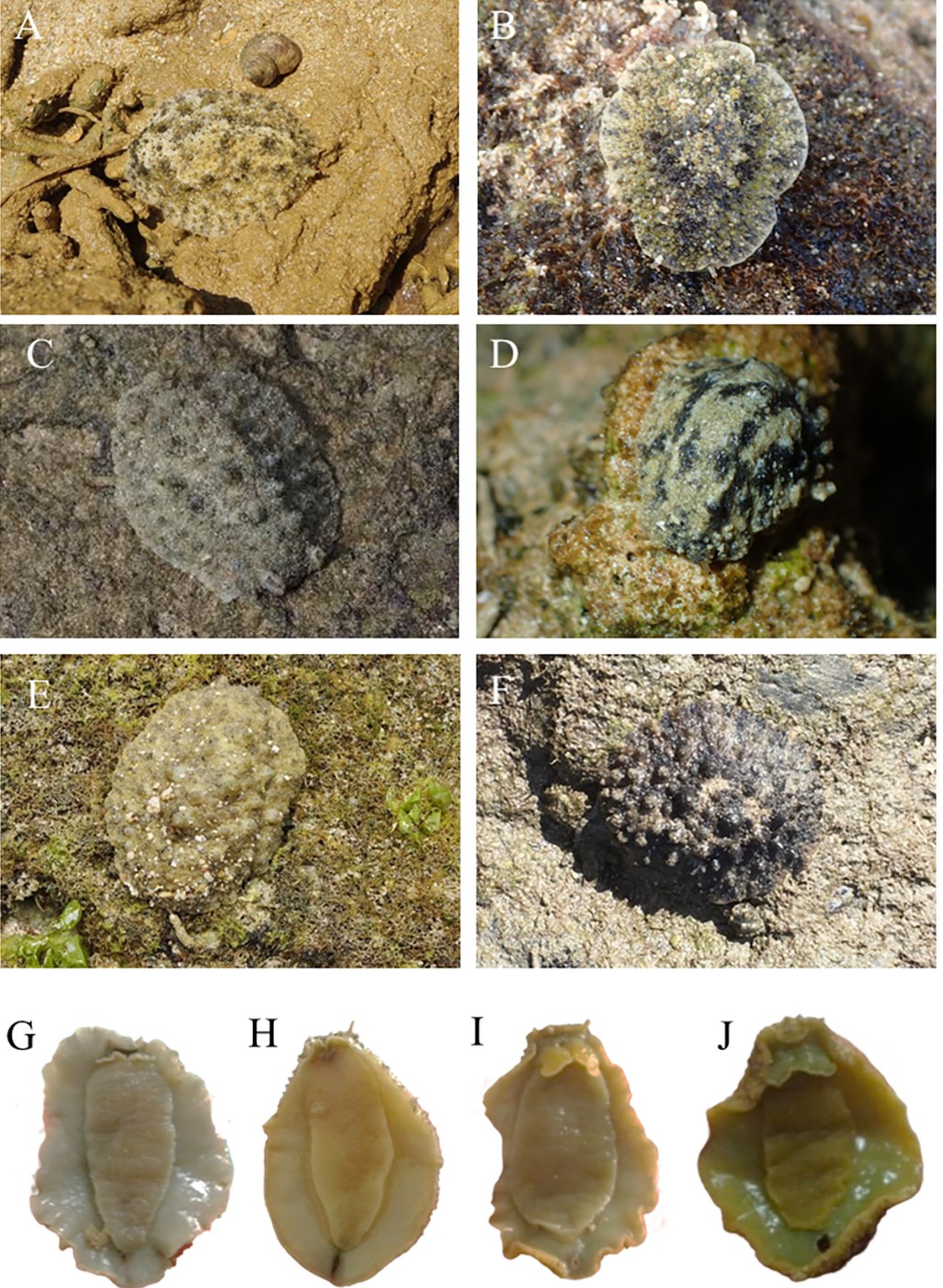

**Figure 7 Live *in situ* photographs of *Peronia verruculata* unit #1.** (A) SR3 (length: 60 mm/width: 25 mm); Okinawajima Island, Seragaki (26°30′25.2″N, 127°51′51.1″E), (B) HD12 (30/20); Okinawajima Island, Hedo, Uza Beach (26°51′57.1″N, 128°15′47.1″E), (C) GS2 (45/32); Okinawajima Island, Gushikawa (26°21′59.4″N, 127°52′24.5″E), (D) IR3 (20/11); Iriomote Island (24°21′56.9″N, 123°55′41.3″E), (E) SNB6 (50/-); Okinawajima Island, Sunabe, Sea Wall (26°19′44.4″N, 127°44′37.8″E), (F) KC1 (61/56); Shikoku, Kochi, Nishidomari (32°46′43.6″N, 132°43′58.3″E). Ventral view of animals. (G) GS2 (45/32), (H) NK3 (47/26); Okinawajima Island, Nakagusuku (26°17′05.4″N, 127°49′00.8″E), (I) TIM2 (50/25); Okinawajima Island, Teima (26°33′06.1″N, 128°03′42.8″E), (J) UKC1 (43/25); Okinawajima Island, Ukachi (26°25′28.4″N, 127°44′43.5″E).

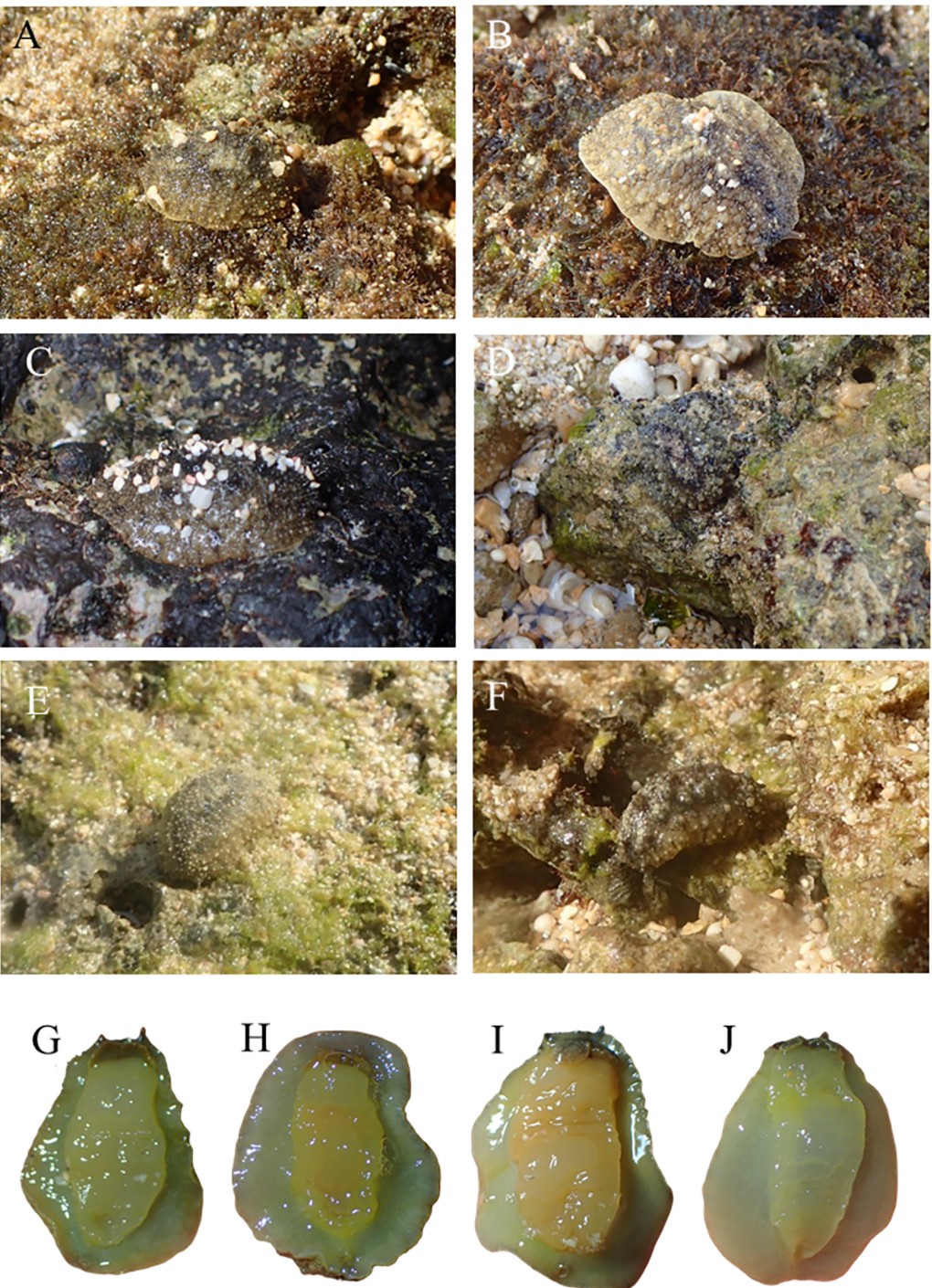

**Figure 8 Live *in situ* photographs of *Peronia setoensis*.** (A) HD10 (length: 20 mm/width: 11 mm); Okinawajima Island, Hedo, Uza Beach (26°51′57.1″N, 128°15′47.1″E) (B) HD11 (20/13), (C) TMR14 (30/16); Amami Oshima Island, Tomori Beach (28°27′43.8″N, 129°43′18.1″E) (D) BAB6 (14/5); Okinawajima Island, Sunabe, Baba Park (26°20′04.9″N, 127°44′35.9″E) (E) BAB10 (16/11), (F) BAB13 (15/8). Ventral view of animals. (G) HD10, (H) HD16 (20/14), (I) TMR2 (27/20), (J) BAB10.

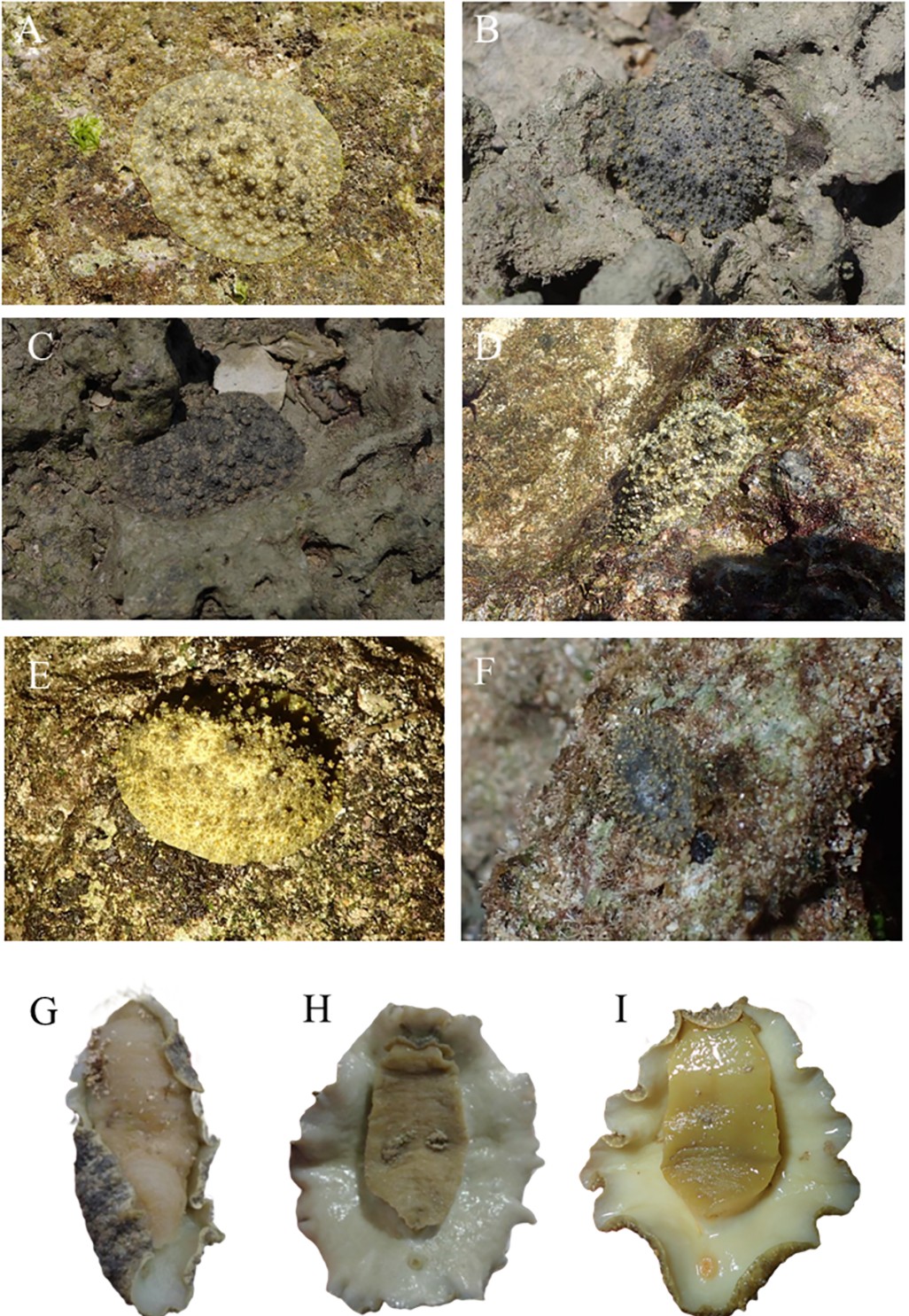

**Figure 9 Live *in situ* photographs of *Peronia peronii*.** (A) SNB5 (length: 110 mm/width: - mm); Okinawajima Island, Sunabe, Sea Wall (26°19′44.4″N, 127°44′37.8″E), (B) NK1 (47/26); Okinawajima Island, Nakagusuku (26°17′05.4″N, 127°49′00.8″E), (C) NK2 (95/70), (D) HD1 (105/73); Okinawajima Island, Hedo, Uza Beach (26°51′57.1″N, 128°15′47.1″E), (E) TMR6 (90/60); Amami Oshima Island, Tomori Beach (28°27′43.8″N, 129°43′18.1″E), (F) TMR13 (14/8). Ventral view of animals. (G) NK1, (H) SNB1 (110/100), (I) TMR6 (90/60).

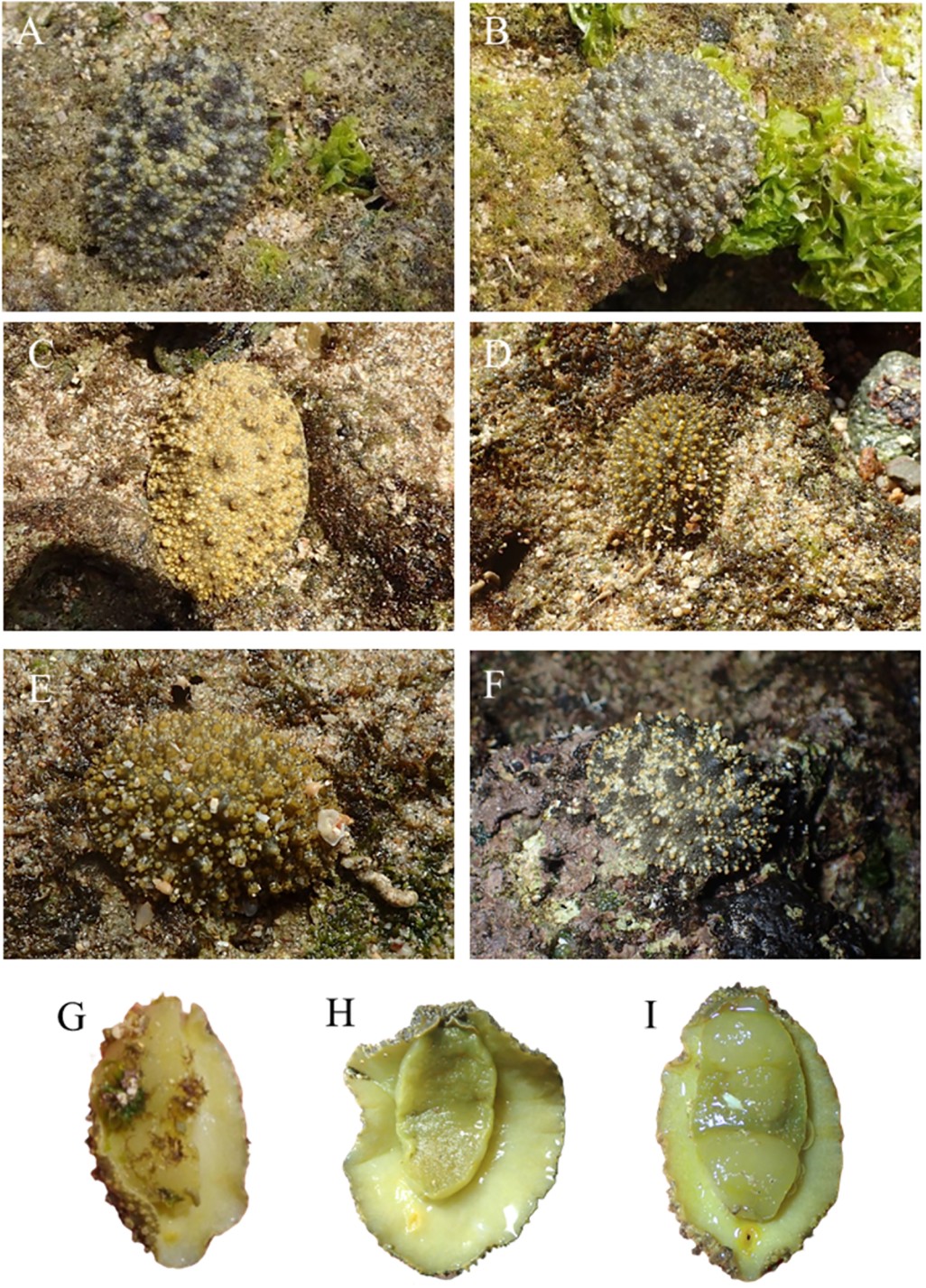

**Figure 10 Live *in situ* photographs of *Peronia okinawensis*.** Goya-like papillae. (A) SNB2 (length: 60 mm/width: 30 mm); Okinawajima Island, Sunabe, Sea Wall (26°19′44.4″N, 127°44′37.8″E), (B) SNB3 (50/-), (C) HD3 (60/40); Okinawajima Island, Hedo, Uza Beach (26°51′57.1″N, 128°15′47.1″E). chestnut shell-like papillae. (D) HD6 (20/13), (E) HD9 (28/18), (F) TMR10 (38/26) Amami Oshima Island, Tomori Beach (28°27′43.8″N, 129°43′18.1″E). Ventral view of animals. (G) SNB3, (H) HD2, (I) TMR8 (60/32).

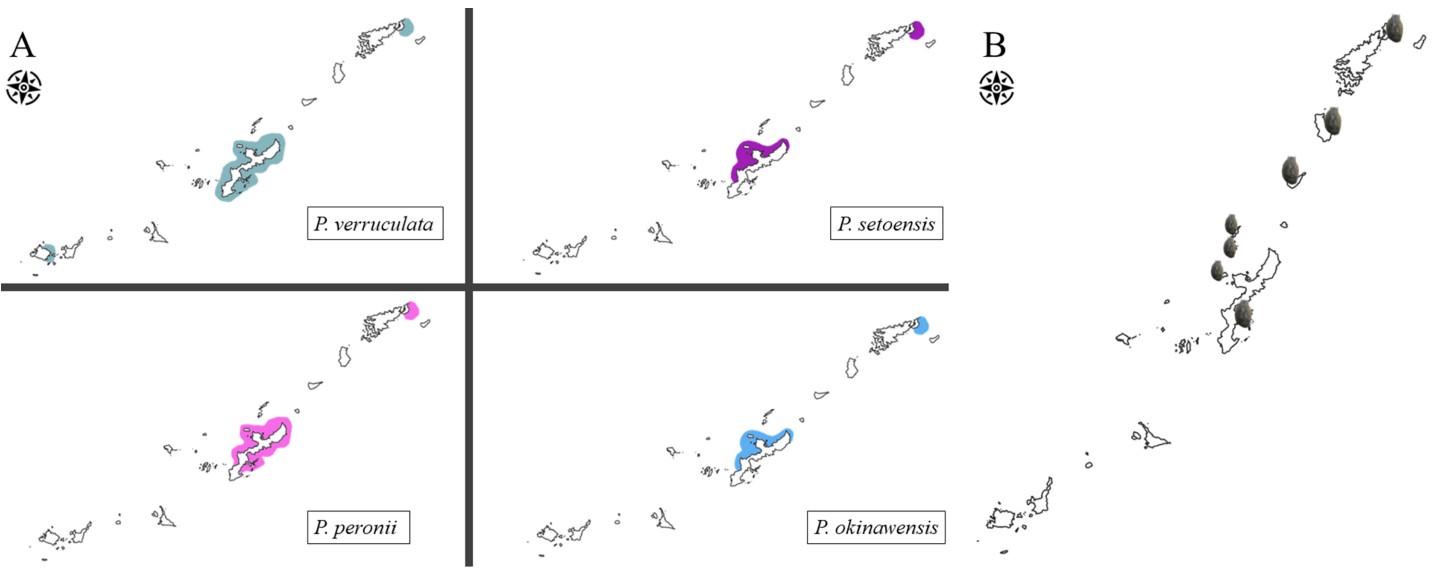

**Figure 11 Map of distribution range of *Peronia* species in the Ryukyu Island.** (A) Distribution ranges of four *Peronia* species in the Ryukyu Islands observed in this study. (B) Locations of local *Peronia* fisheries in the Ryukyu Islands (denoted by small *Peronia*); Amami Oshima, Tokunoshima, Okinoerabu, Iheya, Izena, Ie, and Okinawajima islands (from N to S).

*Peronia* fisheries also exist in China and India (*Solanki, Kanejiya & Gohil, 2017*). A previous study in the Gulf of Khambhat, India, reported local people consume *P. verruculata* due to high levels of proteins, carbohydrates and lipids (*Solanki, Kanejiya & Gohil, 2017*). Local people harvest *P. verruculata* in this area as the species is abundant at local reefs during pre-monsoon to monsoon seasons when other food sources, such as mudskippers, are breeding, and only small animals are available (*Solanki, Kanejiya & Gohil, 2017*). Since our study showed that *P. verruculata* was common around Okinawajima Island and the comparatively larger sizes of the species made them more obvious on reefs, we theorize that local Okinawajima Island fisheries likely primarily catch *P. verruculata*. Moreover, in Amami Oshima Island, *P. okinawensis* was more abundant than *P. verruculata* and it had similar sizes and color variations as *P. verruculata*. Thus, the local fishery in Amami Oshima Island may be harvesting *P. okinawensis*, or it may be that people prefer *P. verruculata* for fisheries and leave *P. okinawensis* and other species in the field. In Japan, *P. verruculata* has been thought to be the only species that exists across Japan, and thus all four species have been generally treated with one Japanese name, "iso awamochi", across Japan. *Peronia* spp. are hard to distinguish from external morphology, so distribution ranges of each species could help achieve a better understanding of *Peronia* fisheries in the Ryukyu Islands. However, further studies are still required to be carried out in more detail around Okinawajima Island as well as around more remote islands, especially at locations currently are known to have *Peronia* fisheries in the Ryukyu Islands, such as Ie, Izena, Iheya, northern Amami Oshima, Tokunoshima and Okinoeraubu islands (Fig. 11).

Previous research has indicated that size differences between Japanese species may have utility for species identification of *Peronia* species, alongside characters of ontogeny

(*Katagiri & Katagiri, 2007*). For example, *Katagiri & Katagiri (2007)* designated *P. setoensis* as the "mini awamochi" as specimens are relatively smaller than "iso awamochi" (*P. verruculata*) (Figs. 5 Ba. and 5Bb.). This current study also found that *P. setoensis* was constantly smaller in both lengths and widths, similar to *Dayrat et al. (2020)*. Since there are only two species of *Peronia* in mainland Japan, in this region identification can possibly reliably depend on size differences. The identification of *Peronia* species by their size has potential implications for their conservation, as fisheries may also target specific sizes of animals (*e.g.*, larger sizes preferred), which then would put increased pressure on larger species. However, in southern Japan, especially in the Ryukyu Islands, based on our results, size differences alone are not a very useful diagnostic for clearly distinguishing all four species. For instance, *Takagi et al. (2019)* did not specifically aim to understand *Peronia* taxonomy, but their study did not recognize more than two species of *Peronia* in southern Japanese waters, and mistakenly identified *P. verruculata* as *P. setoensis*, and larger specimens of *P. peronii* as *P. verruculata* due to the authors' use of size differences only. Although distinguishing the four *Peronia* species from only gross external morphology is almost impossible and problematic (*Ueshima, 2007*), the general characteristics of each species in the Ryukyu Islands found in this study are introduced below. There are limited data on the ecology and developmental biology of *Peronia* species globally, and the current study aimed to provide data on the distribution of co-occurring species in southern Japan waters, and particularly in the Ryukyu Islands. In addition, our updated distribution maps can provide better understanding for local fisheries in these areas, although we clearly need to confirm distributions of *Peronia* spp. around other islands in the Ryukyus. Such distribution mapping will help to inform *Peronia* fisheries in the future.

## Simple summaries of each species

*Peronia verruculata* (Cuvier, 1830) (Figs. 5 Ba. and 7)

Type locality: Red Sea (not described in the original description, but likely the Red Sea according to *Dayrat et al. (2020)*.)

Examined materials: Okinawajima Island: Sunabe, Sea Wall (26°19′44.4″N, 127°44′37.8″E), one specimen, 50 (length)/- (width) mm (SNB6); Sunabe, Baba Park (26°20′04.9″N, 127°44′35.9″E), two specimens, 7/4 mm (BAB0), 32/18 mm (BAB8); Nakagusuku (26°17′05.4″N, 127°49′00.8″E), one specimen, 47/26 mm (NK3); Oujima (26°07′40.4″N 127°46′16.1″E), four specimens, 70/37 mm (OU1), 47/19 mm (OU2), 30/18 mm (OU3), 38/14 mm (OU4); Gushikawa (26°21′59.4″N, 127°52′24.5″E), five specimens, 50/39 mm (GS1), 45/32 mm (GS2), 41/31 mm (GS3), 20/15 mm (GS4), 40/14 mm (GS5); Ukachi (26°25′28.4″N, 127°44′43.5″E), 4 specimens, 43/25 mm (UKC1), 43/25 mm (UKC2), 34/12 mm (UKC3), 40/15 mm (UKC4); Seragaki (26°30′25.2″N, 127°51′51.1″E), five specimens, 30/10 mm (SR1), 34/15 mm (SR2), 60/25 mm (SR3), 45/24 mm (SR4), 20/10 mm (SR5); Teima (26°33′06.1″N, 128°03′42.8″E), three specimens, 45/20 mm (TIM1), 50/25 mm (TIM2), 19/13 mm (TIM3); Hedo, Uza Beach (26°51′57.1″N, 128°15′47.1″E), five specimens, 36/26 mm (HD5), 31/26 mm (HD7), 48/28 mm (HD8), 30/20 mm (HD12), 50/40 mm (HD15). Iriomote Island: southeast of Honera Beach (24°21′56.9″N,

123°55′41.3″E), five specimens, 22/11 mm (IR1), 26/14 mm (IR2), 16/11 mm (IR3), 25/15 mm (IR4), 29/16 mm (IR5). Amami Oshima Island: Tomori Beach (28°27′43.8″N, 129°43′18.1″E), 1 specimen, 36/24 mm (TMR5). Kochi Prefecture: Nishidomari (32°46′43.6″N, 132°43′58.3″E), 2 specimens, 61/56 mm (KC1), 60/52 mm (KC2).

Molecular phylogeny: All *P. verruculata* in this study belonged to phylogenetic unit #1 *sensu* Dayrat et al. (2020). The uncorrelated *p*-COI distances among *P. verruculata* specimens in our study were 0% to 1.6%. Phylogenetic trees showed that *P. verruculata* from the Ryukyu Islands and the West Indo-Pacific did not differ significantly. Moreover, as specimen numbers of *P. verruculata* were the largest among the four species in this study, there were many more haplotypes than observed in other species. One of the haplotypes (Haplotype 1 in COI and 16S) was most common all over Okinawajima Island, as well as in the Ryukyu Islands in 16S analyses, and other haplotypes were low in number. This star-like haplotype pattern indicates that *P. verruculata* may have good dispersal ability, or alternately the species might be a relatively recent arrival to the Ryukyu Islands and southern Japan. As well, the genetic distances between each haplotype were relatively close to each other (min: one base pair change, max: eight base pair changes between each haplotype). For example, even when comparing between Okinawajima Island and mainland Japan, Kochi Prefecture, differences were only three to four base pair changes, further demonstrating a lack of genetic variability within this species.

Morphology: The length of animals varied between a minimum of 7 mm to a maximum of 70 mm (average: 37.56 ± 13.69 mm), which was smaller than the average size of *P. peronii* (see *P. peronii* section). The size ranges of *P. verruculata* were similar to *P. okinawensis*, although *P. verruculata* was slightly smaller in average length. There were significant differences between the animal sizes and collection sites in *P. verruculata*. However, these size differences likely occurred due to low specimen numbers at some sites and to differences in collection month rather than being due to location. For instance, the biggest animal was collected in August and the smallest animal was in December. As well, abundances appeared to differ at the same sites between summer and winter. Thus, analyzing the correlation between animal sizes and collection months in the future may lend further support to significant differences between animal size of *Peronia* species and seasons or collection sites. Furthermore, *P. verruculata* showed the most variation in dorsal color morphotypes among the four species examined in this study. Most *P. verruculata* specimens had a light brown background color and black to green-ish grey mixed or marble pattern. A relatively few *P. verruculata* specimens had a darker body, more similar to *P. okinawensis*. The notum was covered by various numbers of papillae and none of specimens had a smooth notum like *P. setoensis*, but many specimens had relatively less obvious papillae compared to those on *P. peronii* and *P. okinawensis*. The color of papillae was generally the same as the background base body color with a single color to marble patterned specimens. Moreover, some marble-colored specimens had only one color for all papillae. There were also few specimens that had papillae colors different from body color. The ventral side of specimens were yellow-ish cream brown usually with green on the mantle of many specimens. A few specimens had much darker grey colors on their ventral side.

Ecology, distribution, and abundance: *P. verruculata* was the most abundant species in this study and at most study sites, except at Amami Oshima Island where *P. verruculata* was less abundant than the other three species. As well, this species was found from a wide variety of habitats, for example, they were found on artificial structures (*e.g.*, tetrapods), sand, coral rubble, macroalgae, and rocks. Thus, *P. verruculata* might have a high tolerance to different environments, which could explain its wide distribution range and high abundances. Moreover, the color variations of *P. verruculata* could be due to its wide distribution range as well as inhabiting a variety of microhabitats, and they may also be able to adapt their morphotype depending on surrounding microhabitats, although this theory remains to be confirmed. However, for instance, nudibranch sea slugs are known to exhibit different body colorations depending on food sources (*Rudman, 1991*), and perhaps this mechanism occurs within *Peronia* species too. Therefore, in future studies, relationships between color variations of the animals and their microhabitat and diet need to be analyzed, as well as examining gut contents.

Remarks: Some *Peronia* species are known to have a pelagic larval form even though the adults live in the intertidal zone (*Dayrat et al., 2017*; *Dayrat et al., 2020*), and *P. verruculata* is one such species (*Katagiri & Katagiri, 2007*). In the future, more detailed population genetic analyses on *P. verruculata* in the Ryukyu Islands as well as on specimens from mainland Japan to elucidate the dispersal pathways of *P. verruculata* are needed. Since Okinawajima Island contains both relatively pristine to heavily anthropogenically influenced coastlines (*Reimer et al., 2015*; *DiBattista et al., 2020*; *Reimer et al., 2019*), genetic differences may arise between different beaches, as seen in studies on other marine benthic invertebrates (*Hamamoto et al., 2021*). However, our haplotype analyses did not show clear differences depending on location in this study. Expanding collection sites and specimen numbers in the future could help find areas that harbor high genetic diversity and may need protection in the face of ongoing expansion of coastal development in this region (*Masucci & Reimer, 2019*; *Biondi, Masucci & Reimer, 2020*). *Peronia setoensis* Dayrat & Goulding, 2020 (Figs. 5Bb. and 8)

Type locality: Nishimuro, near Seto Marine Biological Laboratory, Wakayama Prefecture, Japan.

Examined materials: Okinawajima Island: Sunabe, Baba Park (26°20′04.9″N, 127°44′35.9″E), 13 specimens, 8 (length)/6 (width) mm (BAB1), 10/7 mm (BAB2), 10/7 mm (BAB3), 7/4 mm (BAB4), 5/4 mm (BAB5), 14/5 mm (BAB6), 8/7 mm (BAB7), 6/4 mm (BAB9), 16/11 mm (BAB10), 7/4 mm (BAB11), 15/9 mm (BAB12), 15/8 mm (BAB13), 13/9 mm (BAB14); Hedo, Uza Beach (26°51′57.1″N, 128°15′47.1″E), 4 specimens, 20/11 mm (HD10), 20/13 mm (HD11), 20/13 mm (HD13), 20/14 mm (HD16). Amami Oshima Island: Tomori Beach (28°27′43.8″N, 129°43′18.1″E), 3 specimens, 24/21 mm (TMR1), 27/10 mm (TMR2), 30/16 mm (TMR14).

Molecular phylogeny: From the phylogenetic analyses, *P. setoensis* was roughly separated into clades depending on collection location. These clades were not only different between the Ryukyu Islands and mainland Japan, but there were also clades for specimens collected in Baba Park and Hedo on Okinawajima Island, and from northern Amami Oshima Island. Moreover, haplotype analyses also showed that each haplotype was

constructed based largely on collection sites, and genetic differences between each haplotype were relatively large (1 to 21 base pair changes; *p*-COI distances 0% to 3.5%). This clear genetic separation depending on collection sites could be due to *P. setoensis* having direct development while *P. verruculata* has planktotrophic development (*Katagiri & Katagiri, 2007*), and thus *P. setoensis*' larval dispersal may be more limited. *Dayrat et al. (2020)* recovered minimum *p*-COI genetic distances between *P. verruculata* unit #1 and *P. verruculata* unit #2 of between 3.2% to 6.6%. Moreover, minimum *p*-COI distances between taxonomically valid species were 4.3% to 5.4 % between *P. willani* and *P. sydneyensis* (*Dayrat et al., 2020*). *Goulding et al. (2021)* mentioned that interpreting meaning from mitochondrial genetic distances of *Peronia* species needs to be carried out with caution as genetic structure could be different between locations due to geographic distances or limited larvae dispersal as a result of ocean current patterns. Thus, in the future, additional specimen collection and DNA barcoding of nuclear genes, such as Histone 3 (H3) and the Internal Transcribed Spacer (ITS), regions are needed to better understand the taxonomy of this species.

Morphology: *Katagiri & Katagiri (2007)* mentioned that the sizes of *P. setoensis* were smaller than *P. verruculata*, and also in the current study we found that *P. setoensis* specimens were consistently smaller compared to sizes of the other three species (largest: 28 mm, smallest: 5 mm in length, average 14.65 ± 7.09 mm). In this study, there were significant differences in animal sizes depending on the collection site, however numbers of specimens greatly differed between locations (Baba Park = 13, Hedo = 4, Amami Oshima Island = 3). Thus, similar to our results for *P. verruculata*, these results cannot be considered as conclusive. Although phylogenetic trees and haplotype analyses' results showed clear separations depending on collection site, the colors of live animals were almost always grey green-ish based color with light brown. Their papillae were generally lighter than the base color, and numbers of dorsal eyes were one to four, and usually three. Dorsal papillae were comparatively less protruding, so the notum of *P. setoensis* was smoother than other *Peronia* species, and in particular when compared to *P. peronii* and *P. okinawensis*. The ventral side was yellow and green-ish in color on the inner side of the mantle.

Ecology, distribution, and abundance: *P. setoensis* had not been reported from the Ryukyu Islands in previous studies, and this report represents the first records of *P. setoensis* from Okinawajima and Amami Oshima islands. The southernmost distribution range of *P. setoensis* until now was Nagasaki Prefecture, Kyushu, mainland Japan, with unconfirmed reports from the Tokara Islands, north of Amami Oshima Island. (*Baba, 1958*; *Dayrat et al., 2020*). In our study, four specimens were found from northern Okinawajima Island at the type locality of *P. okinawensis* in October 2021, as well as three specimens from northern Amami Oshima Island in November 2021, and 13 specimens from the mid-western coast of Okinawajima Island in December 2021. *P. setoensis* were observed in rocky intertidal areas and the coloration and their small sizes made them comparatively cryptic in the field. We found that small *P. setoensis* were often hidden under rocks in tide pools, and thus many individuals may have been missed during our sampling. Since the abundance of *P. setoensis* seemed increase in winter in our surveys,

other sampling locations in the Ryukyu Islands that we investigated in this study may also harbor *P. setoensis*.

Remarks: More specimen collections from both mainland Japan and the Ryukyu Islands as well as further afield (*e.g.*, Hong Kong, Taiwan, South Korea), as well as further analyses *via* both nuclear gene sequence phylogenetic analyses and detailed anatomical studies, are required to more completely understand the distribution and phylogenetic relationships within *P. setoensis*. *P. setoensis* in the Ryukyu Islands may yet be a different species from *P. setoensis* from Wakayama in mainland Japan, but this hypothesis needs to be investigated with caution. *Peronia peronii* (Cuvier, 1804) (Figs. 5Bc. and 9)

Type locality: Mauritius, Indian Ocean.

Examined materials: Okinawajima Island: Sunabe, Sea Wall (26°19′44.4″N, 127°44′37.8″E), three specimens, 110/100 mm (SNB1), 90/- mm (SNB4), 110/- mm (SNB5); Nakagusuku (26°17′05.4″N, 127°49′00.8″E), three specimens, 47/26 mm (NK1), 95/70 mm (NK2), 45/25 mm (NK4); Hedo, Uza Beach (26°51′57.1″N, 128°15′47.1″E), one specimen, 105/81 mm (HD1). Amami Oshima Island: Tomori Beach (28°27′43.8″N, 129°43′18.1″E), three specimens, 90/60 mm (TMR6), 80/60 mm (TMR7), 14/8 mm (TMR13).

Molecular phylogeny: The genetic differences of all *P. peronii* specimens collected in this study were relatively small (uncorrelated *p*-COI distances = 0% to 1.1%), and they all were in the same clade. This clade also contained sequences retrieved from GenBank from specimens collected from the West Indo-Pacific region (Papua New Guinea, Guam, Okinawajima Island). However, a GenBank sequence from Mauritius was different from the specimens from the Ryukyu Islands as well as from those of the West Indo-Pacific, indicating *P. peronii* in the West Indo-Pacific may possibly be a different species than *P. peronii* in Mauritius, the type locality of this species. Furthermore, haplotype analyses indicated that one common haplotype (haplotype COI: 24, 16S: 13) was found all over the Ryukyu Islands, with several less common haplotypes.

Morphology: *P. peronii* is known to be comparatively large in size and this study confirmed this, with the largest *P. peronii* specimen reaching 110 mm in length (smallest: 14 mm, average: 78.6 ± 32.49 mm). This was the largest individual among all *Peronia* specimens in this study. Adult *P. peronii* were easy to distinguish from other species in the field due to their large sizes, but small *P. peronii* looked similar to *P. okinawensis* (Fig. 10F). The color patterns on the dorsal side of *P. peronii* were varied to plain light brown to dark brown, and some specimens had both colors mixed. From photograph analyses, the general color of *P. peronii* was much darker than *P. verruculata*, although the types of color variations were similar. Papillae colors of *P. peronii* were usually the same as base body colors, and they had big yet sporadically located papillae and tiny inconspicuous papillae between large papillae on their notum. The ventral side was white and yellow-ish cream brown to darker brown, and some specimens had green-ish colors as observed in other species.

Ecology, distribution, and abundance: *P. peronii* has a unique distribution range within the Ryukyu Islands. According to *Dayrat et al. (2020)*, the distribution range of *P. peronii* in Japan was limited to Okinawajima Island and islands around Amami Oshima Island.

The current study also supported that *P. peronii* is observed around the Ryukyu Islands, as we did not record this species from Iriomote Island, in the Yaeyama Islands. The previous northernmost record of *P. peronii* was from the Tokara Islands (*Baba, 1958*), and our study confirmed the distribution reached at least to northern Amami Oshima Island. Other islands of the Ryukyus may harbor *P. peronii*, for example more northern Yakushima and Tanegashima islands, and this remains to be confirmed in the future. Despite the wide distribution range of *P. peronii*, the number of specimens collected was the smallest among the four species in this study, which may be due to this species possibly being nocturnal (*Dayrat et al., 2020*). In this study, night sampling was carried out just once at Tomori Beach, Amami Oshima Island, during which we found three *P. peronii* specimens, and also specimens of all three other *Peronia* species. More nocturnal observations need to be carried out to confirm the activity rhythm of *P. peronii* in the Ryukyu Islands.

Remarks: Previous studies have mentioned that *P. verruculata* is less active during the night, at least around Okinawajima Island (*Hamaguchi & Yoshioka, 2002*). Their study was carried out in August 20 to September 4, 1999, when the lowest tides were observed during the daytime rather than at night, and they found that activity times of *P. verruculata* were correlated with the length of the lowest tide (*Hamaguchi & Yoshioka, 2002*). However, the lowest tide is observed at night in winter in the Ryukyu Islands, indicating that the activity rhythm of *Peronia* species might be different depending on the season as low tides vary between daytime in summer to nighttime in winter. Thus, not only examining whether *P. peronii* is a nocturnal species, but also differences in species compositions and species abundances depending on the time and season of collection remain to be examined in future work. *Peronia okinawensis* Dayrat & Goulding, 2020 (Figs. 5Bd. and 10)

Type locality: South East of Cape Hedo (Uza Beach), Okinawajima Island.

Examined materials: Okinawajima Island: Sunabe, Sea Wall (26°19′44.4″N, 127°44′37.8″E), two specimens, 65 (length)/- (width) mm (SNB2), 50/- mm (SNB3); Hedo, Uza Beach (26°51′57.1″N, 128°15′47.1″E), five specimens, 60/35 mm (HD2), 60/40 mm (HD3), 45/35 mm (HD4), 20/13 mm (HD6), 28/18 mm (HD9). Amami Oshima Island: Tomori Beach (28°27′43.8″N, 129°43′18.1″E), six specimens, 39/25 mm (TMR4), 60/32 mm (TMR8), 33/22 mm (TMR9), 40/20 mm (TMR10), 38/26 mm (TMR11), 50/31 mm (TMR12).

Molecular phylogeny: All *P. okinawensis* specimens collected in this study as well as the holotype from Hedo (sequences from GenBank) were in the same single clade. The molecular analyses of *P. okinawensis* showed the smallest intraspecific genetic differences among the four *Peronia* species we examined (uncorrelated *p*-COI distances: 0% to 0.9%). Haplotype analyses also showed the smallest haplotype numbers among four species for both COI and 16S sequences (COI: 5, 16S: 3 haplotypes). There was one common haplotype found from both Okinawajima Island and Amami Oshima Island (haplotype COI: 30, 16S: 18), and a few haplotypes from Amami Oshima Island and Hedo were observed with one to three base pair changes.

Morphology: *P. okinawensis* had a length similar to *P. verruculata*, but was slightly bigger than *P. verruculata* (largest: 65 mm, smallest: 20 mm, average: 45.23 ± 13.82 mm). The numbers of collected specimens of *P. verruculata* were about three times more than

those of *P. okinawensis* in this study, and we believe the size ranges likely are almost the same between these two species. The dorsal color patterns of *P. okinawensis* were generally darker than those seen in *P. verruculata* and much green-ish in general. Moreover, similar to *P. peronii*, the notum of *P. okinawensis* was covered with spiky, or more protruding papillae, which was often obvious in the field. The shapes of papillae among our specimens were roughly classified into "goya"-like papillae (Figs. 10A–10C) and chestnut shell-like papillae (Figs. 10D–10F). The average lengths for specimens with goya-like papillae were 55.71 ± 7.32 mm (largest: 65 mm, smallest: 45 mm, *n* = 7), and 33 ± 7.8 mm (largest: 40 mm, smallest: 20 mm, *n* = 6) for specimens with chestnut shell-like papillae. The number of papillae with dorsal eyes per specimen were slightly different between two morphotypes (goya-like papillae: 21–26; chestnut shell-like papillae: 18–23), as well as the numbers of dorsal eyes on per papillae (goya-like papillae: 1–5; chestnut shell-like papillae: 3–5). There are tiny papillae with no dorsal eyes were sporadically covered between the papillae with dorsal eyes in both morphotypes. These tiny papillae were about half size of papillae with dorsal eyes for goya-like papillae and much smaller for chestnut shell-like papillae. The color of papillae was lighter than body color. In the field, the goya-like shape of each papilla appeared relatively large and more densely located than seen in *P. peronii*. On the other hand, chestnut shell-like papillae were much more dense and were tiny, located all over the notum. The chestnut shell-like papillae were observed in *P. verruculata* unit #1 specimens from Bali in *Dayrat et al. (2020)*, but were not found in this study. These differences in external morphology might be due to animal sizes and their developmental stages, but this needs to be investigated further.

Ecology, distribution, and abundance: The results of this study expand the distribution range of *P. okinawensis*, and also include the first records of this species from Amami Oshima Island. These results support *Dayrat et al. (2020)*, who mentioned that *P. okinawensis* was possibly not endemic to Okinawajima Island and may be distributed in other parts of the Ryukyu Islands. In Okinawajima Island, there was a clear pattern with *P. okinawensis* distributed only on the western coast of the island. Originally, we thought that *P. okinawensis* only inhabited beaches with comparatively less anthropogenic impacts, as the only three specimens found in 2004 were from a pristine location in northern Okinawajima Island. However, in this study, we found this species not only from relatively pristine northern Okinawajima Island and Amami Oshima islands, but also from moderately disturbed beaches, such as at Sunabe, on the mid-western coast of Okinawajima Island. Moreover, *P. okinawensis* was not reported in the Yaeyama Islands, similar to *P. peronii*, and thus the current southern limit of this species is Okinawajima Island. Moreover, there is the possibility that *P. okinawensis* is also a nocturnal species like *P. peronii*. Night specimen collection was only carried out in Amami Oshima Island, and Amami Oshima Island recorded the highest abundance of *P. okinawensis*, and this idea needs to be confirmed in future studies.

Remarks: This study reports new distribution records of *P. okinawensis* from Amami Oshima Island, supporting the idea that this species is not endemic to only Okinawajima Island. However, as there were few haplotypes and small genetic distances observed, and *P. okinawensis* may be endemic to the Ryukyu Islands or a comparatively small area.

Further sampling collection is needed within the Ryukyu Islands, especially around the Sakishima Islands, as well as in southern mainland Japan, to more clearly confirm the distribution of this species.

## CONCLUSIONS

This study focused on genus *Peronia* in the Ryukyu Islands, and reassessed the number of species present while mapping their distributions. We discovered that four species exist in the Ryukyu Islands, including new and southernmost records of *P. setoensis*, new records of *P. okinawensis* in Amami Oshima Island, and reconfirming the distribution of *P. peronii* in Amami Oshima Island, which represents the second-northernmost record for this species. In future studies, increasing the number of specimens from mainland Japan and from other islands of the Ryukyus are required to determine more precisely the distribution ranges of *Peronia* species in this region. It is hoped that the current study's data facilitates a better understanding of genus *Peronia* diversity in the Ryukyu Islands and southern Japan waters, and that the generated data can start to inform fisheries management.

## ACKNOWLEDGEMENTS

The authors thank all Molecular Invertebrate Systematics and Ecology Laboratory (MISE) members at the University of the Ryukyus (UR), particularly Dr. Yuka Kushida and Kairi Takahashi for help in specimen collection. We also thank Dr. Euichi Hirose and Dr. Tohru Naruse (UR) for checking an earlier version of this work.

### Funding

This work was supported by KAKENHI Fostering Joint International Research (B) and Iori Mizukami was supported by a Okinawa Research Core for Highly Innovative Discipline Science (ORCHIDS) scholarship and Ryudai Kagakuin at University of Ryukyus. The funders had no role in study design, data collection and analysis, decision to publish, or preparation of the manuscript.

### Grant Disclosures

The following grant information was disclosed by the authors:
KAKENHI Fostering Joint International Research.
Okinawa Research Core for Highly Innovative Discipline Science (ORCHIDS) Scholarship.
Ryudai Kagakuin at University of Ryukyus.

### Competing Interests

James D. Reimer is an Academic Editor for PeerJ.

## Author Contributions

- Iori Mizukami conceived and designed the experiments, performed the experiments, analyzed the data, prepared figures and/or tables, authored or reviewed drafts of the article, and approved the final draft.
- Chloé Julie Loïs Fourreau conceived and designed the experiments, performed the experiments, analyzed the data, authored or reviewed drafts of the article, and approved the final draft.
- Sakine Matsuo performed the experiments, authored or reviewed drafts of the article, and approved the final draft.
- James Davis Reimer analyzed the data, authored or reviewed drafts of the article, and approved the final draft.

## Field Study Permissions

The following information was supplied relating to field study approvals (*i.e.*, approving body and any reference numbers):

Kagoshima Prefecture permit Shirei Oshima Rinsui/April 2021–March 2022.

## DNA Deposition

The following information was supplied regarding the deposition of DNA sequences:

The Peronia sequences newly obtained in this study are available at GenBank: ON241849 to ON241929, ON241987 to ON242057, ON243660 to ON243661, and ON243753 to ON243757.

## Data Availability

The trimmed concatenated sequences (COI + 16S) and length and width of Peronia spp are available in the Supplemental Files.

## Supplemental Information

Supplemental information for this article can be found online at http://dx.doi.org/10.7717/peerj.13720#supplemental-information.

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
