# Peer review of "Diversity and distribution of air-breathing sea slug genus Peronia Fleming, 1822 (Gastropoda: Onchidiidae) in southern Japanese waters"

_PeerJ, doi:10.7717/peerj.13720_

## Round 0.1 · original submission · Minor Revisions

I have received reports back from two expert reviewers and their comments can be seen below. Overall, both have favourable comments about the manuscript, but they have also raised a number of minor, but nevertheless important issue that need to be taken care of in a revised version of the manuscript. Please ensure that you attend all of these comments as I am sure they will improve the quality of the manuscript.

Reviewer 1 ·

Basic reporting

The language is ok, though reads clunky in some parts.
The flow is ok.
I have checked and confirm that GenBank accession numbers are live.

Can I check if the specimens used in their study have been or will be deposited at any natural history collections? If so, please state, and if not, perhaps include contact information to someone responsible for these specimens? Should another researcher wish to review them at a later date.

Line 180–183: Sentences could do with a bit of re-ordering; the present flow is confusing. Suggest to (1) mention that you did ABGD first, then (2) mention you used the online portal, and the settings used. Finally, (3) mention that you used the same COI dataset. To add on, authors could also clarify why it was written as “COI trees from ML analyses”. As far as I recall, ABGD does not require an input tree (vis-à-vis Poisson Tree Process, see Zhang et al., 2013). I am unable to confirm on my end, as the link provided in the manuscript does not work at time of writing this review.

Fig. 4A: Missing a label for P. setoensis.

Figs. 5–7: Figs. 6 & 7 were mentioned first before Fig. 5 in the manuscript (Line 250 vs. Line 258), so there is some ordering issue here. I think Figs. 5 & 6 can be combined into one figure tile since they both convey the same message on size differences between species (the boxplots are quantitative while the dorsal images are more qualitative), so one can become tile A and the other tile B of the newly combined figure.

Fig 7. Suggest that authors include significant pairwise comparisons mentioned in Lines 260–263 into the respective boxplots for better clarity.

Experimental design

I have one issue regarding their analysis. May I know why the authors chose to perform ABGD analysis with K2P model? But calculate uncorrected p-distances for COI (mentioned Line 223)? Most studies I have seen generally do one or the other. Please also see the critique by Srivathsan & Meier (2012) on the inappropriate use of the K2P model in DNA barcoding literature. I suggest sticking to the use of uncorrected p-distances instead. Link: https://onlinelibrary.wiley.com/doi/full/10.1111/j.1096-0031.2011.00370.x

Lines 178–179: Was there a specific package in R that you used to construct a pairwise distance matrix? Do include it here. Relatedly, authors also mentioned later on having calculated uncorrected p-distances for COI. How was that calculated? I find no mention of that in the Methods, please add it in.

Validity of the findings

Given that Onchidiidae is a poorly-studied family, the authors study of Peronia here will be useful in adding to the extant knowledge of sea slugs.

I have some comments on some things mentioned by the authors:

Line 30: Authors mentioned that they “identified a potential species complex within P. setoensis” in the abstract. This claim needs to be tempered, as they do not provide justifications why this is so. It was mentioned that genetic variation was large (Line 229) but it would be good to give some sort of yardstick for which they base their judgements on. Are they applying the 3% threshold here to delimit species? In addition, a 3.6% variation is within range of known intraspecific distances in P. verruculata, and below interspecific ranges of 4.3–5.4% between P. williani and P. sydneyensis. Both examples they described (Lines 432–435) do not provide compelling evidence to suggest potential cryptic speciation in P. setoensis; they just point to a need for more careful interpretation of mitochondrial genetic variation results in the genus.

Line 488: “…could be a different species”? You cautioned careful interpretation of results earlier for P. setoensis. Please exercise the same caution here! Your sequence GB18 originated from Dayrat et al. (2020). Their study combined morphology and genetic data (both mitochondrial and nuclear) and yet, they did not erect the Mauritius clade as a new species. Furthermore, your node in Fig. 4 is not even well-supported to begin with. I cannot stress this enough—genetically different does not always mean a different species!

Additional comments

Title: Perhaps consider removing “of species” from the title, to become “Diversity and distribution of the air-breathing sea slug genus Peronia Fleming, 1822 in southern Japanese waters”. Authors could also perhaps include some of the higher taxon classifications here to provide more context, e.g., (Gastropoda: Onchidiidae).

Lines 72: Is there a citation for the phrase “fisheries may not completely understand which species they are harvesting”? Or do the authors mean that it is actually researchers who do not have data/information on what species are harvested in the Peronia trade?

Line 95: Change “a DNA barcoding technique” to “DNA barcoding”.

Line 171: repetitions

Line 213: P. verruculata unit #1 (sensu Dayrat et al., 2020).

Line 295: Can emphasise that P. setoensis was previously only known from Wakayama, Honshu, Japan.

Line 318: Change “(e.g., larger individuals being targeted)” to “(i.e., larger sizes preferred)”.

Line 353: 0 to 1.7%. Standardize with Lines 228–229.

Line 301: Authors should avoid speculating which fisheries capture which Peronia spp based species distributions. Being in the same locality as a naturally-occurring Peronia population does not necessarily mean those are the species that will be harvested. Supply-chains? Fisheries could be importing their onchs from elsewhere. Does size really matter? What about taste? I think there is just not enough information on the Peronia trade, or context provided, to make this link.

Line 358: The haplotype pattern is referred to as “star-like”

Line 439 or 469: Authors could mention DNA barcoding of nuclear genes, or genome sub-sampling as potential ways to better understand genetic variation in P. setoensis.

Line 483: See comment on Line 353.

Reviewer 2 ·

Basic reporting

In this manuscript, the authors sampled Peronia species in the Ryukyu Islands to better understand the geographic distributions of these intertidal slugs in the waters of southern Japan, and investigate possible external morphological differences. This study was based on a DNA barcoding approach and using phylogenetic analyses to cluster new DNA sequences with identified DNA sequences from a previous taxonomic study by Dayrat et al. (2020). The authors expanded the known distributions of multiple species, and provided photographs of live specimens in situ for the recently described species P. setoensis. This is a well-written paper that I enjoyed reading, that includes relevant references to recent literature on onchidiid diversity and previous studies on Peronia species in Japan. There were a few references missing from the Literature Cited, the authors should add the details for their references to Katagiri 1998 (Line 64), Johnson & Gosliner (Line 147), and Edgar 2004 (Line 152). Also note that on line 180, the Puillandre reference is listed as published in 2011 while in Literature Cited it is listed as 2012. The manuscript also includes background information on the fishery of Peronia species in Japan and how these organisms are prepared, which I found quite interesting and is relevant to future work to understand how harvesting may impact the relative abundance of these species locally.

In the results section, the ML analysis of the concatenated dataset is not described (lines 207-208). How did the topology of that analysis differ?

All of the figures are relevant, and overall there are high-quality. I especially like Figure 6 with the photos of the 4 species of Peronia that approximate the average size of each species, and the plates with in situ photographs as well as the high quality images of the ventral coloration. I also appreciate that a map was included (Fig. 12B) showing the locations of Peronia fisheries. There are some minor corrections of the figures or captions needed that I have noted here:
In Figure 4A, the haplotypes belonging to P. setoensis are not labeled with the species name.
For Figures 8 & 9, the captions indicate that the ventral view of the animal is labeled A-D, but in the figure they are labeled G-J. Similarly, for Figures 10 & 11, the captions indicate that the ventral view of the animal is labeled A, B, and C but in the figure they are labeled G, H, and I.
In Figure 12A, it is confusing to have the map of the islands repeated 4 times very closely together. I strongly recommend revising Figure 12A to place the island groups for each species distribution map further apart so that the map for each species can be clearly differentiated.

I recommend adding a brief figure caption to the supplementary tables to clarify the content to the readers (particularly to label which Table gives GenBank accession numbers for COI sequences, and which is for the 16S sequences). Also note that some of the supplementary tables are referred to with numbers (S Table 1) and others with S preceding the number (S Tables S2 & S3).
Some of the figure numbers need to be corrected in the text. Throughout the morphology section (including headings on line 331 for P. verruculata, lines 412, 474, 528) the figure numbers are incorrectly reported. For P. verruculata the figures are Fig. 6A and Fig 8, not 7A & Fig. 9. Please review if the figure references in the text are correct.
One last suggestion on the figures, in Figure 1, the most northern locality is labeled Nishidomari, but in the charts in Figure 7 this locality is referred to by the Prefecture name Kochi. It would be easier for the reader if the same name was used consistently for this locality in all figures and the main text (Kochi is used in the text, so I suggest updating the name in Figure 1 to Kochi or “Nishidomari, Kochi”).

Experimental design

The research question in this study is well-defined and provides data to improve the understanding of species distributions of these marine slugs in Japan. It expands on previous research on Peronia species with expanded sampling and DNA barcoding efforts. The methods are described in sufficient detail. The specific primer sequences provided in Table 1 are not strictly necessary as these primers are previously published and widely used and cited, and are cited in the Materials and Methods section. Table 1 could be omitted without negatively impacting the paper.

Validity of the findings

The phylogenetic analyses and haplotype network analyses are appropriate and well-described. There is some inconsistency between the haplotype network illustrated in Figure 4 and the description in the results. Based on Figure 4, it looks like the text in Line 243-244 should read 6 to 14 base pairs changes in COI and 1 to 5 base pair changes in 16S (or if the text is correct, the figure needs to be corrected).

There are some issues in reporting of the morphological data:
Lines 247-250: the smallest spm of P. verruculata is listed as 16/11mm but in the Supp. Table 1, the specimen BAB0 is listed as being 7/4 mm. Also note that the average length given here (38± 12.94 mm) is not the same as listed in the morphology section of P. verruculata (37.56 ± 13.69 mm). I observed the same inconsistency in averages reported in P. setoensis between the Results and Morphology sections, so these numbers need to be checked for all species to ensure the correct values are being reported. Similarly, the largest specimen of P. setoensisis is listed as 28/16 but I only see a specimen listed as 30/16. I request that the authors review their data and correct the parts of the text that are not consistent with their measurements.

The statistical analyses are appropriate.

Additional comments

The paper is well-organized and provides observations from the field on the abundance and ecology of each species. The species P. setoensis had previously only been described from a single site, so the additional data on its distribution in southern Japan is fascinating. At the beginning of the Discussion, the authors mention that this is the only region with four Peronia species found. This is very interesting and makes the Peronia in Japan a very interesting group of onchidiids for further study! I think it would be worth mentioning in the abstract that this is the only island group where four Peronia species have been found to co-occur. Readers will naturally wonder how these Peronia species differ from closely related species in other regions, where only one or two species are found. I suggest to the authors to indicate in the Discussion that there is limited data on the ecology and developmental biology of Peronia species globally, but this study aimed to provide data on the distribution of co-occurring species in Japan with notes on their abundance and habitats. Below I detail a few areas that I think need clarification, a few minor edits and some suggestions. Overall, I think this paper contributes to our knowledge of onchidiid geographic distributions, and I recommend it for publication with minor revisions.

Line 56- I haven’t seen the word tetrapods used before as a substrate, I have only seen this word to refer to a clade of animals. I see later in the manuscript that you are using this term to refer to artificial structures. If this is the correct terminology, I would suggest adding “artificial structures” to this line to clarify its meaning, as you have done on line 390.
Lines 202-203: Add to the end of this sentence that the new DNA sequences generated for this study were deposited in GenBank: “From a total of 87 specimens, 81 COI and 75 16S sequences were successfully sequenced, and in total 84 specimens were identified to species level. All new COI and 16S sequences generated for this study are deposited in GenBank (Table S1).”
Line 230- change “I examined” to “examined” since this study has multiple co-authors
Line 270, elsewhere: if the scientific name is at the beginning of a sentence, instead of abbreviating the genus name as P. it should be written out as Peronia.
For line 322- This sentence is confusing to me. It says that Takagi et al. 2019 mistakenly identified specimens of P. verruculata as P. setoensis, but P. setoensis had not yet named and formally described in 2019 and was not mentioned in Takagi’s paper. In the Takagi paper, the majority of Peronia specimens sequenced were correctly identified as P. cf. verruculata and the specimens identified as Peronia sp. (Grp 2) belong to P. peronii. So, I don’t see that any specimens of Peronia in the paper were misidentified. If you disagree, be specific about which specimens were misidentified and the evidence that they are misidentified. In the introduction to that paper, Takagi referenced previous work that suggested a small hidden species of Peronia that they referred to as Peronia sp. (= P. setoensis) but it was not sampled in Takagi’s study.
Line 400: Dayrat et al. 2020 indicated that the larvae of onchidiids are released in sea water, but this does not necessarily mean that all Peronia species have pelagic larvae, some species may crawl away. I recommend qualifying this statement at the beginning of the sentence. For example, it could be edited to read: “At least some Peronia species have a pelagic larval form, but the reproductive mode differs between some Peronia species (Ueshima, 2007) and needs to be investigated further.”
In the section on the Molecular phylogeny for P. setoensis it is mentioned that there are genetic differences between specimens collected from different localities. I think this is an interesting finding. It is up to the authors, but they could add a statement in this section to highlight that this species was previously only known from a single locality, and that the expanded sampling in this study is the first to reveal genetic structure within the species. As you noted, the clear correlation of the genetic structure observed with the sampling localities would be expected with direct larval development. It would be interesting if future research investigated differences in larval development and ecology further between these Peronia species.
Line 431: Is it Katagiri & Katagiri, 2007 that indicated developmental differences between the Peronia species or Ueshima, 2007?
Line 566-567: There are many statements throughout the manuscript of things that need to be studied in the future, but this is normal in scientific research. Perhaps his text could be combined with the previous sentence to make it more concise. Something like “These differences in external morphology might be due to animal sizes and their developmental stages, but this would need to be investigated further."
Line 575-576: Interesting observations!

---

## Round 0.2 · accepted · Accept

I am satisfied with the comments made to the manuscript.